

**Shortwave Radiative Impacts of the Asian Tropopause Aerosol Layer (ATAL)**
**using Balloon-borne In-situ measurements at three distinct locations in India**
Vadassery Neelamana Santhosh[1] , Bomidi Lakshmi Madhavan[1], Sivan Thankamani Akhil
Raj[2], Madineni Venkat Ratnam[1], Jean-Paul Vernier[3,4], and Frank Gunther Wienhold[5]
[1]National Atmospheric Research Laboratory (NARL), Gadanki 517 112, India
[2]India Meteorological Department (IMD), New Delhi 110 003, India
[3]National Institute of Aerospace, Hampton, VA, USA
[4]NASA Langley Research Center, USA
[5]Institute of Atmospheric and Climate Science (IAC), ETH, Zurich, Switzerland
**Correspondence to**:
Bomidi Lakshmi Madhavan (madhavanbomidi@gmail.com, blmadhavan@narl.gov.in)
**Abstract**
The recurring presence of the Asian Tropopause Aerosol Layer (ATAL) in the Upper
Troposphere Lower Stratosphere (UTLS) region, strongly linked with the Asian Summer Monsoon
Anticyclone (ASMA), has garnered significant attention over the past decade. However, despite
advances in instrumentation, studies quantifying the radiative impacts of ATAL aerosols in terms
of radiative forcing and heating rates remain limited. This study aims to address this gap by
evaluating the direct radiative effects of ATAL aerosols in the UTLS using in-situ measurements
from the Balloon measurement of the Asian Tropopause Aerosol Layer (BATAL) campaigns
conducted between 2014 and 2019 over three distinct locations in India: Gadanki (13.48°N,
79.18°E), Hyderabad (17.47°N, 78.58°E), and Varanasi (25.27°N, 82.99°E). The study considers
three scenarios where UTLS aerosols are predominantly composed of sulfates, nitrates, or
anthropogenic aerosols. Our findings reveal significant changes in aerosol radiative forcing,





ranging from -0.015 to 0.03 Wm$^{-2}$ at the top of the atmosphere, -0.01 Wm$^{-2}$ to -0.16 Wm$^{-2}$ at the
surface, and 0 to 0.19 Wm$^{-2}$ within the atmospheric column when transitioning from sulfate to
nitrate and anthropogenic aerosol scenarios. UTLS aerosols were found to contribute 0.1% to 2.3%
of the total columnar atmospheric forcing, with the highest contributions observed under the
anthropogenic scenario. Notably, heating rate profiles indicate enhanced aerosol heating under
anthropogenic scenarios, with rates reaching up to 0.03 K day$^{-1}$, particularly over Varanasi,
compared to significantly lower rates under sulfate and nitrate scenarios. The study highlights the
spatial variability in radiative impacts across different locations, reflecting the structural and
dynamic complexities of ATAL within the ASMA region. It emphasizes the need for a
comprehensive approach combining in-situ, satellite, and model-based retrievals to overcome
current limitations and achieve a more accurate understanding of the net radiative impacts of
ATAL aerosols.
**Keywords:** *Asian Tropopause Aerosol Layer; Aerosol-Radiation Interaction; Radiative forcing*
*and heating rates; Upper Troposphere-Lower Stratosphere*
**1. Introduction**

The Asian summer monsoon (June-August) over the northern hemisphere is known for

transporting pollutant-laden air masses over a vast geographic region. A large-scale anti-cyclonic
circulation, known as the Asian Summer Monsoon Anticyclone (ASMA), forms in the Upper
Troposphere Lower Stratosphere (UTLS) due to intense heating over the Tibetan plateau coupled
with persistent deep convection over the head Bay of Bengal (BoB). This system traps and isolates
air masses, dispersing them across a broad geographic area (10º N to 40º N and 10º E to 140º E)
(e.g. Park et al., 2007; Randel and Park, 2006), leading to persistent extremes of trace constituents
such as water vapor, methane, nitrogen dioxide, ozone, etc. around the ASMA center (e.g. Basha



et al., 2021; Kumar and Ratnam, 2021; Park et al., 2007). Satellite observations have also revealed

a recurrent layer of aerosol enhancements, known as the Asian Tropopause Aerosol Layer

(ATAL), in the UTLS region (~ 13-18 km), which significantly impacts stratospheric composition,

chemistry, cirrus cloud characteristics, and the Earth's radiative balance (Vernier et al., 2011;

2015; Thomason and Vernier, 2013).

The formation and dissipation of ATAL are closely linked to deep convection during the

monsoon, which transports aerosols from the Bay of Bengal and the surrounding land areas into

the UTLS region (He et al., 2020). Aerosols are non-homogeneously distributed within the ATAL,

and descending motion in the western part of the ASMA region plays an important role in the

dissipation of the layer. Fadnavis et al. (2013) demonstrated through simulations that the deep

convection and the associated heat-driven circulation over the southern side of the Himalayas are

the dominant transport pathway of aerosols into the UTLS together with notable anthropogenic

contribution. Neely et al. (2014) further emphasized broader source regions for ATAL aerosols

beyond solely Asian $SO_2$ emissions. Specifically, their results showed that $SO_2$ emissions from

China and India contributed ~30% of the sulfate aerosol extinction in the ATAL during

volcanically quiescent periods. Recent simulations using GEOS-Chem indicated that the

contribution from India and China could double those estimates, with both countries contributing

equally (30%) (Fairlie et al., 2020). Lau et al. (2018) indicated two preferred pathways for the

strong vertical transport of the carbonaceous and dust aerosols toward the ATAL region, one over

the Himalaya-Gangetic Plain (India) and the other one over the Sichuan basin (China) located in

the southern and eastern foothills of the Tibetan Plateau respectively. These different sources and

pathways indicate the possible presence of wide ranges of natural and anthropogenic aerosols,




which are further influenced by the dynamic and chemical processes, including secondary aerosol
formation, within the ASMA region as suggested by recent chemical analysis (Appel et al., 2022).
There are varied opinions on the chemical composition of the ATAL region. Vernier et al.
(2015) noted that the bottom part of the ATAL is dominated by the sulfate aerosols, with the
carbon-to-sulfate ratio ranging from 2 to 10. Later, model simulations by Yu et al. (2015) revealed
a dominant sulfate contribution together with surface-emitted and secondary organics. Fadnavis et
al. (2013) also reported the presence of sulfate aerosols together with black carbon (BC), organic
carbon (OC), and mineral dust. The significant lofting of mineral dust to the ATAL region was
noted in the simulations by Ma et al. (2019), and they further revealed that hygroscopic aerosols
(such as nitrates and sulfates) and associated liquid water influence the extinction in the UTLS
region. Lau et al. (2018) previously reported the presence of dust, carbonaceous aerosols, and
carbon monoxide (CO), which are lofted through orographically forced deep convection into the
ATAL region. Long-term simulations (2000 to 2015) by Bossolasco et al. (2021) indicated that
aerosols other than mineral dust in the ATAL consist of ~ 40% sulfate, 30% secondary aerosols,
15% primary aerosols, 14% of ammonia-based aerosols, and less than 3% BC. GEOS-Chem
simulation by Fairlie et al. (2020) further revealed a dominant contribution of nitrate aerosols at
the southern side of the ASMA region. While the findings from model-based studies varied, many
in-situ studies concur on the relative dominance of nitrate aerosols in the ATAL. The first-ever
offline chemical analysis from the BATAL campaigns over India revealed that there exists a
dominant nitrate contribution in the ATAL, and surprisingly, the sulfate aerosols were below the
detection limit of the instrument (Vernier et al. 2018). Later, Vernier et al. (2022) indicated the
dominant presence of nitrate and nitrite aerosols with concentrations between 88 and 374 ng m$^{-3}$
at STP during the 2017 BATAL campaign. Höpfner et al. (2019) demonstrated the dominant





ammonium nitrate particles in the upper troposphere during the Asian monsoon period using
satellite and high-altitude aircraft measurements combined with atmospheric trajectory
simulations and cloud-chamber experiments. Appel et al. (2022) observed enhancements in the
mass concentrations of particulate nitrate, ammonium, and organics in altitudes between ~ 13 and
18 km using airborne instruments. Their aerosol mass spectrometry analysis further revealed that
the particles in the ATAL mainly consist of ammonium nitrate (AN) and organics.
Owing to the complexity of retrieving the aerosol properties required for the radiative
impact estimations, the studies on ATAL radiative forcing and heating rates are sparse. Vernier et
al. (2015) used long-term satellite measurements to determine that the summertime aerosol optical
depth over Asia, associated with the ATAL, increased from 0.002 to 0.006 between 1995 and
2013. This increase resulted in a short-term regional forcing at the top of the atmosphere of -0.01
$Wm^{-2}$, compensating for about one-third of the radiative forcing associated with the global increase
in $CO_2$. They also noted that the regional radiative forcing caused by the ATAL varies between
clear-sky and all-sky conditions. Under all-sky conditions, calculations showed lesser shortwave
radiative forcing over the monsoon region due to cloudiness. Using simulations with MERRA-2
reanalysis data, Gao et al. (2023) demonstrated that ATAL impacts clear-sky shortwave fluxes at
the TOA and surface. For the time-averaged ATAL relative to the no-aerosol case, the net effects
include a 0.15 W $m^{-2}$ increase in incoming solar radiation at the TOA and a 0.72 W $m^{-2}$ reduction
in absorbed shortwave radiation at the surface. Over the past decade, the radiative forcing due to
ATAL led to a summertime reduction in surface temperature, although this effect has not yet been
quantified.
Despite the recognized importance of ATAL in climate studies, research on its radiative
forcing and heating rates from in-situ measurements is non-existent due to challenges in retrieving



relevant aerosol properties. The Balloon measurement campaigns for the Asian Tropopause
Aerosol Layer (BATAL) consist of high-resolution in-situ measurements of aerosol and
atmospheric properties at three different locations over India. This study aims to address several
research aspects, including the extent of aerosol enhancement in the UTLS during the late
monsoon, the radiative forcing across different scenarios of UTLS aerosols, and the resulting
heating rate patterns.
In the following sections, we provide a brief description of the BATAL campaigns and study
locations (**Section 2**), describe the datasets used (**Section 3**), and details of the methodology for
estimating radiative impacts (**Section 4**). The results are discussed in **Section 5**, followed by the
listing of key findings in **Section 6**.
**2. Campaign Details and Observation Sites**
The Balloon Measurement Campaigns of the Asian Tropopause Aerosol Layer (BATAL)
was conducted jointly by the Indian Space Research Organization (ISRO) and the National
Aeronautics and Space Administration (NASA) during the last phase of the monsoon season (July
to September) from 2014 to 2019. These campaigns involved over a hundred balloon flights
equipped with miniature payloads to study the optical properties, size distribution, and
composition of aerosols in the ATAL. The BATAL also focused on investigating ozone and water
vapor behavior in the UTLS and the impact of deep convection over the ATAL region. For more
detailed information on the payloads, balloon types, and other scientific objectives, refer to Vernier
et al. (2018). These experiments were conducted in three distinct locations (in terms of the local
weather and surface emissions) in India (**Fig. 1**):



(i)    **Gadanki (13.48°N, 79.18°E)**: A rural background location in southern peninsular India
with hilly topography. The site experiences surface emissions primarily from vehicular
sources, agricultural activities, and wood burning. During the monsoon season, surface
pressure ranges from 960 to 965 hPa, with temperatures between 27°C and 30°C. The
prevailing south-westerly winds range from 1.5 to 1.6 m/s, and the relative humidity is
typically less than 60%. The Aerosol Optical Depth (AOD) at 500 nm ranges from 0.4
to 0.5, dominated by coarse-mode aerosols (Santhosh et al., 2024a; Madhavan et al.,
2021).

(ii)    **Hyderabad (17.47°N, 78.58°E)**: A rapidly urbanizing megacity located on the Deccan
Plateau. It has a semi-arid climate, with significant seasonal variations in temperature
and humidity. Local emissions are highly polluted, with long-range aerosol transport
prevalent during the monsoon. AOD measurements during the monsoon season
indicate a dominant contribution from coarse-mode aerosols (Ratnam et al., 2020;
Sinha et al., 2012).
(iii)    **Varanasi (25.27°N, 82.99°E)**: An urban location in the Indo-Gangetic Plain (IGP),
experiencing a humid subtropical climate with significant seasonal variations in
temperature and rainfall. The region is highly polluted, with large variability in aerosol
loading observed throughout the year. Coarse-mode aerosols, primarily dust, dominate
during the pre-monsoon months, while fine-mode anthropogenic aerosols are more
prevalent during post-monsoon and winter months (Murari et al., 2017; Tiwari and
Singh, 2013).
**3. Datasets**



We used data from radiosondes, ozonesondes, and the Compact Optical Backscatter
Aerosol Detector (COBALD) from the BATAL to derive the aerosol extinction and atmospheric
parameter profiles. In this study, we used in-situ measurements covering a minimum altitude of 20
km. COBALD measurements after August 15, 2017, were excluded to avoid the possible influence
of the Canadian wildfires and the Raikoke eruption (Akhil Raj et al., 2022). The details of the data
used in this study are provided in **Table 1**.

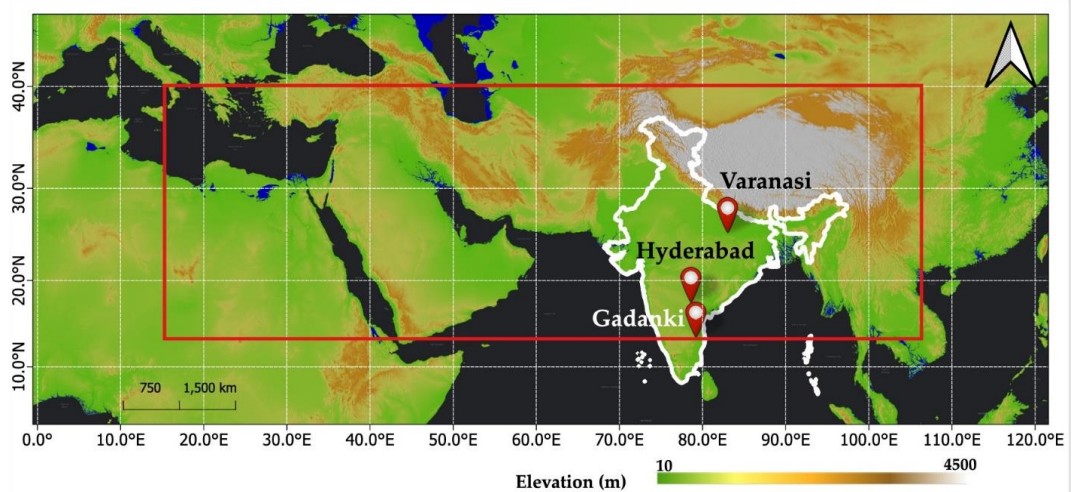

**Figure 1:** The balloon-launching locations during the 2014-2018 BATAL campaigns. The region enclosed in the red color box is the typical geographic extent of the ATAL region (15ºE to 105ºE; 15º N to 40º N)

**Table 1***: The details of balloon launches and the payloads used (COBALD, Ozonesonde, and Radiosonde) in this study have a minimum of 20 km altitude coverage from the surface. The letter 'Y' denotes if the data from a particular payload is available. Meanwhile, the letter 'N' denotes non-availability. All launches are conducted in the local night-time (UTC + 05:30)

| Location | Date (DD-MM-YYYY) and Time (UTC) of launch | COBALD | OZONE SONDE | RADIO SONDE |
|---|---|---|---|---|
| Gadanki (13.48°N, 79.18°E) | 18-08-2014, 15:00 | Y | N | Y |
| | 19-08-2014, 15:30 | Y | Y | Y |
| | 07-09-2016, 19:50 | Y | Y | Y |



| | | | | |
|---|---|---|---|---|
| | 09-09-2016, 15:00 | Y | Y | Y |
| | 31-07-2017, 18:00 | Y | Y | Y |
| | 01-08-2017, 18:00 | Y | Y | Y |
| **Hyderabad (17.47°N, 78.58°E)** | 01-08-2015, 17:00 | Y | N | Y |
| | 05-08-2015, 22:00 | Y | Y | Y |
| | 06-08-2015, 22:00 | Y | Y | Y |
| | 08-08-2015, 18:00 | Y | N | Y |
| | 09-08-2015, 22:00 | Y | N | Y |
| | 13-08-2015, 18:00 | Y | Y | Y |
| | 08-08-2018, 21:00 | Y | Y | Y |
| | 17-08-2018, 20:00 | Y | Y | Y |
| | 26-08-2018, 20:00 | Y | Y | Y |
| | 28-08-2018, 20:20 | Y | Y | Y |
| **Varanasi (25.27°N, 82.99°E)** | 22-08-2015, 18:00 | Y | N | Y |
| | 22-08-2015, 22:00 | Y | N | Y |
| | 04-08-2016, 23:00 | Y | N | Y |
| | 06-08-2016, 21:00 | Y | Y | Y |
| | 08-08-2016, 21:30 | Y | Y | Y |

*3.1. Radiosonde and Ozonesonde*

We utilized pressure, temperature, and relative humidity (RH) data from radiosondes and

ozone volume mixing ratios from ozonesondes. The Meisei (RS-11 G) and iMet radiosondes were
used to measure temperature and pressure at different altitudes. The iMet radiosondes used piezo-
resistors for atmospheric pressure measurements with an accuracy of 1–2 hPa. The Meisei
radiosonde, which lacks a pressure sensor, calculated pressure using temperature and GPS altitude





data. The ozone profile was obtained using EN-SCI Electrochemical Concentration Cell (ECC)
ozonesondes, following the method by Komhyr et al. (1995).  More details on these methodologies
are available in Ratnam et al. (2014) and Akhil Raj et al. (2015).

Relative humidity and ozone mixing ratios were converted to absolute densities using

equations described by Santhosh et al. (2024a). **Fig. 2** shows the extracted mean profiles of
pressure, temperature, water vapor density, and ozone density over the entire study period across
the locations. We found that the temperature, water vapour density, and ozone density
measurements vary across the locations, while the differences in the pressure measurements are
negligible. We have seen the range of water vapour measurements over Varanasi at below
boundary layer (altitude < 2 km), free troposphere (2 to 12 km) and UTLS (12 to 20 km) altitudes
are higher (13 to 21 g m$^{-3}$, 60 mg m$^{-3}$ to 13 g m$^{-3}$, and 0.4 to 60 mg m$^{-3}$, respectively) in comparison
with Gadanki (12 to 19 g m$^{-3}$, 30 mg m$^{-3}$ to 11.7 g m$^{-3}$, and 0.9 to 30 mg m$^{-3}$, respectively) and
Hyderabad (11.3 to 17 g m$^{-3}$, 44 mg m$^{-3}$ to 10.75 g m$^{-3}$, and 0.4 to 40 mg m$^{-3}$, respectively). Similar
way, the ozone measurements within the boundary layer, free troposphere, and UTLS were also
higher over Varanasi (0 to 61 µg m$^{-3}$, 34 µg m$^{-3}$ to 61 µg m$^{-3}$, and 34 to 215 µg m$^{-3}$, respectively)
in comparison with Gadanki (0 to 50 µg m$^{-3}$, 30 µg m$^{-3}$ to 50 µg m$^{-3}$, and 30 to 160 µg m$^{-3}$,
respectively) and Hyderabad (5.3 to 39 µg m$^{-3}$, 28.2 µg m$^{-3}$ to 39.1 µg m$^{-3}$, and 29 to 176 µg m$^{-3}$,
respectively). Regarding temperature, Varanasi profiles were warmer than the other locations,
especially in the UTLS region (208.9 to 230 K) compared to Hyderabad (207.12 to 227.3 K) and
Gadanki (205.3 to 226 K).

We also assessed the biases in these measurements by comparing them with data from the

Microwave Limb Sounder (MLS), Atmospheric Infra-Red Sounder (AIRS), and Modern-Era
Retrospective analysis for Research and Applications, Version 2 (MERRA-2). The biases ranged





from -5 to 5 hPa (-1.5 to 1.5 %) for pressure, -5 to 5 K (-3 to 3%) for temperature, -2 to 2 gm$^{-3}$ (-
150 to 150%) for water vapor density, and -50 to 40 (-100 to 40%) µg m$^{-3}$ for ozone density. It
must be noted that the satellite and reanalysis measurements show higher biases at higher altitudes,
and this is discussed in detail by Santhosh et al. (2024a), which makes the in-situ measurements
more reliable at these altitudes

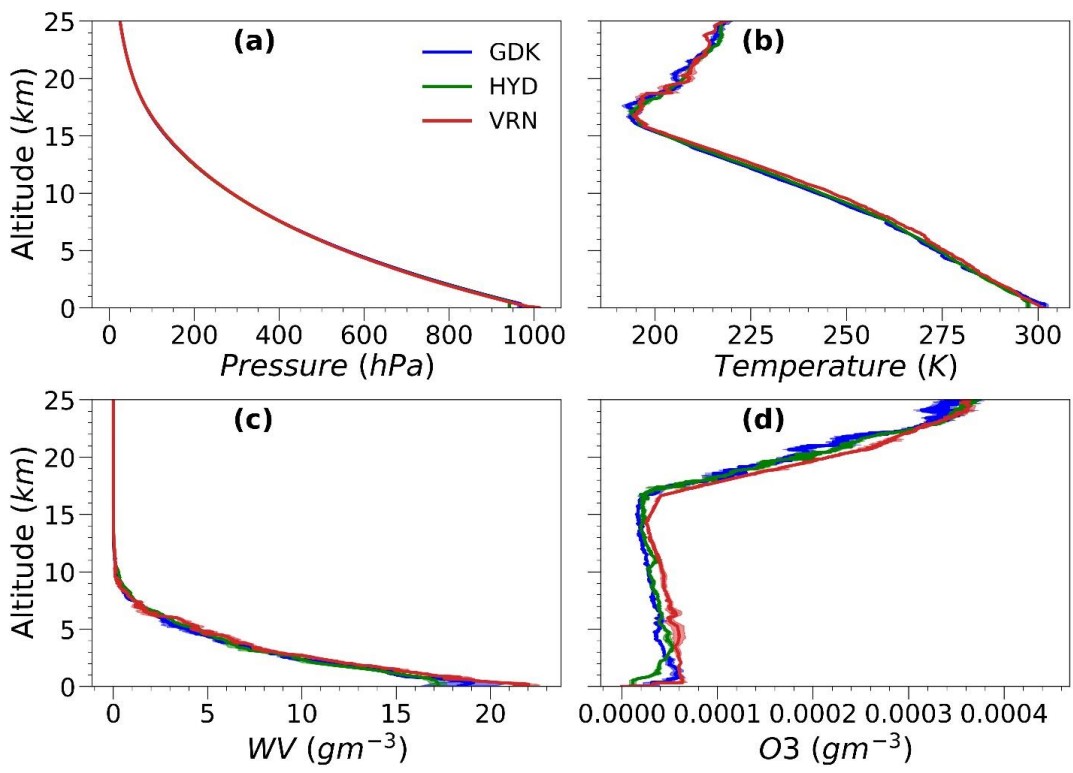


**Figure 2:** The mean profiles of (a) pressure, (b) temperature, (c) water vapor density (WV), and (d) ozone density (O3) with ±1σ standard errors across the study locations.

***3.2. Compact Optical Backscatter AerosoL Detector (COBALD)***

COBALD is a lightweight balloon-borne sonde developed by ETH Zurich that measures

backscattered light from aerosols, molecules, and clouds. This sonde was designed for nighttime



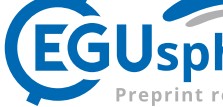

operation only and uses two LED light sources emitting at 455 nm and 940 nm wavelengths. It
detects backscattered light from particles up to 10 meters away, with a precision better than 1% in
the UTLS region (Ravi Kiran et al., 2022; Vernier et al., 2015; 2018). Data are transmitted in real-
time along with pressure and temperature readings at a frequency of 1 Hz.
**3.3. Other Ancillary Datasets**
**3.3.1. MERRA-2 Reanalysis Data**
We used the Modern-Era Retrospective analysis for Research and Applications, Version 2
(MERRA-2), Hourly, Time-averaged, Single-Level, Assimilation, Aerosol Diagnostics 0.625° X
0.5° V5.12.4 (M2T1NXAER) product developed by NASA's Global Modelling and Assimilation
Office (GMAO) for retrieving the AOD for the campaign period. This dataset was chosen as there
were no collocated nighttime retrievals of AOD from in-situ measurements. Further, Che et al.
(2019) reported the performance of the MERRA-2 AOD measurements is better in the South Asian
region (correlation coefficient, r = 0.84; root-mean-square error, RMSE = 0.18; mean absolute
error, MAE = 0.11 and mean fractional error, MFE = 34.54%) based on the comparison with
AERONET observations on a global scale making it a suitable alternative when the ground-based
retrievals are unavailable. It provides total aerosol optical depth (AOD) at 550 nm and the
Angstrom exponent (AE) in the 470-870 nm region. The AODs at 455 and 940 nm wavelengths
were derived using the Angstrom power law (Angstrom, 1964) given by
$$AOD_{\lambda_0} = AOD_{550} \left( \frac{\lambda_0}{550} \right)^{-AE} \qquad (1)$$

where $\lambda_0$ represents the required wavelength. As our required wavelengths are just outside the
wavelength range of the Angstrom exponent (-25 nm in the blue region and +70 nm in the red
region), calculations beyond this range assume that the same power-law relationship holds. Across





the study locations, Varanasi had the highest mean AOD ($0.37 \pm 0.13$), followed by Hyderabad
($0.28 \pm 0.05$) and Gadanki ($0.26 \pm 0.07$). The Angstrom exponent (AE) was highest in Varanasi,
indicating a greater fine-mode aerosol contribution, while Hyderabad had the lowest AE,
suggesting coarse-mode dominance. The AE over Gadanki indicated a more complex aerosol mix.
We have identified the most likely aerosol types in the boundary layer and free troposphere over
all three locations using the cluster analysis of the seven days of air mass back trajectories at 500
m and 4000 m a.m.s.l with HYbrid Single Particle Lagrangian Integrated Trajectory (HYSPLIT)
model (Draxler & Hess, 1998).
*3.3.2. Moderate Resolution Imaging Spectroradiometer (MODIS)*

Surface reflectance is a critical parameter in estimating the aerosol radiative forcing. We

used the MCD43A4 Nadir Bidirectional Reflectance Distribution Function (BRDF)-Adjusted
Reflectance (NBAR) product. This product (MODIS/Terra Nadir BRDF-Adjusted Reflectance
Daily L3 Global 500m SIN Grid) provides reflectance for each MODIS spectral band (centered at
0.469, 0.555, 0.645, 0.859, 1.24, 1.64, 2.13 μm) at local solar noon
(https://lpdaac.usgs.gov/product/mcd43a4v061). We found similar surface reflectance values
across the locations in the visible spectrum, with some deviations in the infrared region, where
Hyderabad had the highest reflectance and Varanasi the lowest (**Fig. S1**).
**4. Methodology**

A schematic of the methodology used in this study is shown in **Fig. 3,** and the steps

involved are discussed in the subsections
*4.1. In-situ aerosol extinction from COBALD measurements*

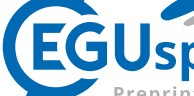



As COBALD measurements capture the total backscattered light from a mix of aerosols,
clouds, and molecules, isolating aerosol contribution from the total backscatter will be difficult
when clouds are present, necessitating the identification and exclusion of these in-situ
measurements. The total backscatter signal is typically expressed in terms of backscatter ratio
(BSR) given by the following equation
$$BSR = \frac{\beta_{Total}}{\beta_{mol}} \qquad (2)$$
where $\beta_{mol}$ represents the molecular backscatter coefficient, whereas the $\beta_{Total}$ includes both
particle and molecular contribution (in the absence of clouds). The contribution from molecular
Rayleigh scattering is determined using the radiosonde's simultaneous temperature and pressure
recordings. The color index (CI) is defined as the 940-to-455nm ratio of the aerosol component of
the BSR given by
$$CI = \frac{BSR_{940} - 1}{BSR_{455} - 1} \qquad (3)$$
CI, being independent of particle number concentration, is a useful metric for interpreting
the particle size. Both BSR and CI serve as indicators of the presence of aerosols. For example,
cirrus clouds can be detected either separately from blue and red channel BSR measurements or
by taking advantage of the CI, enabling distinct discrimination between ice particles (CI < 7) and
aerosol (CI > 7), as highlighted in Hanumanthu et al., (2020). Considering these aspects, we used
$BSR_{455} < 1.12$ for the blue channel measurements, similar to Akhil Raj et al. (2022) and set
$BSR940 < 2.5$ and CI > 7 for the red channel following Vernier et al. (2015) for screening the
aerosols in the UTLS (above 10 km).



Since the above cloud screening criteria are not yet validated for screening aerosols in the
troposphere, we used an approach combining both the vertical gradients of air temperature and
relative humidity (RH) and the altitude-dependent thresholds of RH to determine the clouds in the
lower and free troposphere as described in Xu et al. (2023). Note that we restricted this cloud
screening method below 10 km owing to the high uncertainties associated with radiosonde
measurements in probing RH beyond this altitude. A brief description of this method is provided
in the supplement (**Section S1**), along with an example (**Fig. S2**).
The molecular backscatter coefficients were determined using the temperature and pressure
profiles obtained from the radiosonde measurements (Collis and Russel, 1976)
$$\beta_{mol,\lambda}(z) = \frac{P(z)}{R_d T(z) M} \left(\frac{\lambda}{550}\right)^{-4.09} \times 10^{-32} m^{-1} sr^{-1} \quad (7),$$

where $\lambda$ is the given wavelength, $R_d = 287$ J $K^{-1}kg^{-1}$ is the gas constant for the dry air, and M =
$4.81 \times 10^{-32}$ kg is the molecular weight of dry air expressed in kilograms. Using this, the aerosol
backscatter coefficients are obtained below:
$$\beta_{aer,\lambda}(z) = \beta_{mol,\lambda}(z)(BSR(z) - 1) \quad (8)$$

The aerosol backscatter coefficients were then multiplied with a lidar ratio of 40 sr to get
the extinction coefficient profiles. This particular ratio was used by several studies over the Indian
region for deriving the vertical extinction profiles from the backscatter profiles (e.g. Gupta et
al.2021). Further, to overcome the limitation due to uncertainties in lidar ratios, the aerosol
extinction coefficient profiles ($\beta_{ext}$ (z)) at a given wavelength '$\lambda$' have been normalized using the
MERRA-2 AOD (at '$\lambda$') as
$$\beta_{ext,scaled}(\lambda,z) = \beta_{ext}(\lambda,z) \times \frac{AOD_{MERRA-2}(\lambda)}{AOD_{COBALD}(\lambda)} \quad (9)$$



where $AOD_{COBALD}(\lambda)$ represents the AOD obtained by integrating the derived extinction
profiles at the given wavelength. This scaling also ensures consistency between the columnar
loading and the extinction profiles obtained which eventually helps to reduce the biases in the ARF
and HR estimates (Santhosh et al., 2024b).

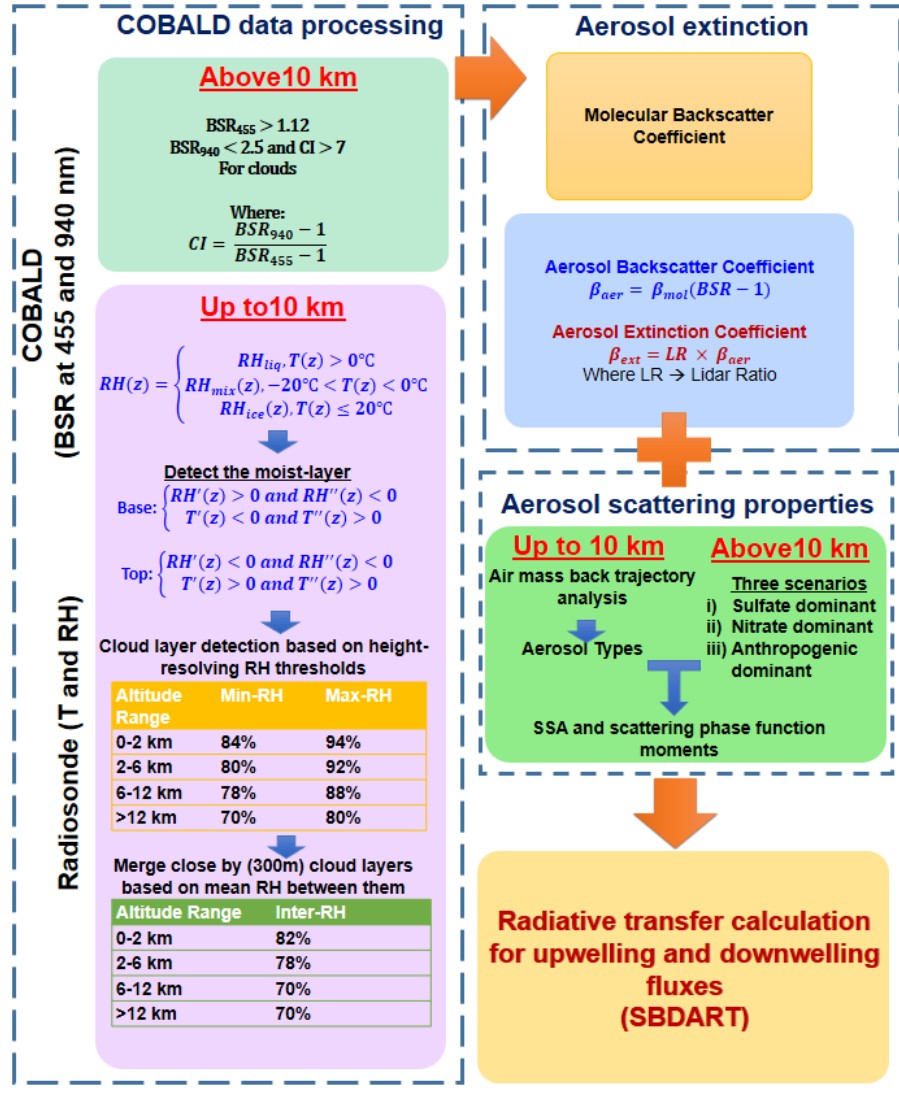


**Figure 3:** The layout of the methodology used in this study



*4.2. Inferring aerosol types for scattering properties*
The balloon-borne measurements of Single Scattering Albedo (SSA) and Asymmetry
parameter (ASY) are not possible. So, we adopted the following strategies for the possible aerosol
types in the UTLS and below (within the boundary layer and free troposphere).
The aerosol composition of ATAL in the UTLS region remains uncertain and inconsistent.
Earlier studies reported the presence of sulfates (Fadnavis et al., 2013; Li et al., 2005; Ma et al.,
2019; Hopfner et al., 2019; Bossolasco et al. 2021), nitrates (Vernier et al. 2018; Hopfner et al.
2019; Ma et al. 2019; Fairlie et al. 2020; Vernier et al. 2022; Yu et al., 2022), and absorbing
aerosols of anthropogenic origin transported from the lower troposphere (Li et al. 2005; Fadnavis
et al. 2013; Lau et al. 2018; Bossolasco et al. 2021). Offline chemical analysis from BATAL
revealed a dominant nitrate contribution (Vernier et al. 2018). Appel et al. (2022) argued that
ATAL consisted solely of secondary substances, namely an internal mixture of nitrate, ammonium,
sulfate, and organic matter. Considering all together, we assumed three different scenarios of
aerosols, say, sulfates, nitrates, and minimum to non-absorbing mixed-type, are dominant in the
UTLS region.

(a) A Sulfate aerosol model consisting of 75% $H_2SO_4$ serves as a typical background

aerosol (Hess et al., 1998).

(b) Nitrate aerosol model (Zhang et al., 2012) with dominant accumulation mode following

a log-normal size distribution (mode radius:0.15 μm, standard deviation: 1.9). This

assumption is consistent with the inferences from the StratoClim field campaigns

(Mahnke et al., 2021), where they noticed the main mode of the aerosol size distribution

shifts towards the accumulation mode with increase in the altitude from beneath the

lower edge of ATAL. Further, they detected the vertical particle mixing ratio within





the ATAL ~700 mg$^{-1}$ for the particles in the size range 65 nm to 1µm and a higher
mixing ratio (>2500 mg$^{-1}$) for the particles whose diameters are larger than 10 nm. They
also noticed that the particles below the ATAL are influenced by the nucleation of
aerosol particles (diameter < 65 nm).
(c) The minimum absorbing mixed-type aerosol model includes the anthropogenic
contribution and is represented by the continental clean type of aerosols (Hess et al.,
1998). This aerosol type encompasses continental regions with minimal to no
anthropogenic influence, typically containing less than 0.1 µg m$^{-3}$ of soot. Its
composition consists of slightly dominant water-soluble aerosols (59%) and insoluble
aerosols (41%). The water-soluble part of aerosol particles originates from gas-to-
particle conversion and consists of sulfates, nitrates, organic, and other water-soluble
substances. Thus, it can be used to describe anthropogenic aerosol, which is just beyond
the sulfates. The water-insoluble part of aerosol particles, on the other hand, consists
of a certain amount of organic material together with soil particles. Thus, despite two
of our study locations (Hyderabad and Varanasi) being heavily urbanized with a high
likelihood of anthropogenic emissions lifting towards the UTLS, the assumption of a
continental clean aerosol model sets a baseline for anthropogenic aerosols with minimal
absorption in the solar spectral range. Hereafter, we denote the continental clean as
'ANTH' as it's a baseline for the anthropogenic/absorbing type of aerosols. Earlier,
Gadhavi and Jayaraman (2006) also used this aerosol model together with sulfate
aerosol models in the stratosphere to estimate the aerosol radiative forcing over
Hyderabad.



The RH-specific SSA and ASY values of Sulfate (SUL), Nitrate (NIT), and Anthropogenic
(ANTH) aerosols at wavelengths of 455 nm and 940 nm (**Fig.4**) were assigned to corresponding
altitude bins based on their respective RH levels to obtain the profiles of SSA and ASY in the
UTLS.

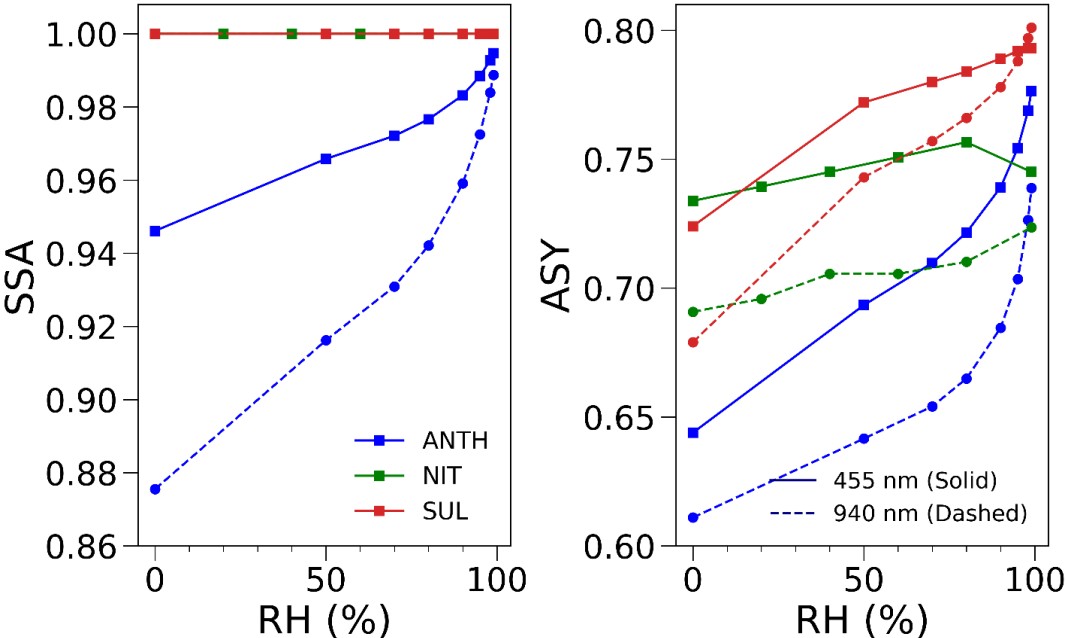


**Figure 4**: The variation of SSA and ASY for the three dominant categories of UTLS, namely anthropogenic (ANTH), nitrate (NIT), and sulfate (SUL) at different RH bins.

To determine the likely aerosol types within the boundary layer (WBL) and free
troposphere (FT), we used the clustered mean seven-day air mass back trajectories at 500 m and
4000 m a.m.s.l to represent the region within the boundary layer and free-troposphere,
respectively, over the study locations using HYSPLIT (Stein et al., 2015) The clustering has been
done for every air mass for the campaign period at a given location. The aerosol type classification
in this way mainly depends on the air mass origins, region of transport, the residence time of air





masses in a particular area, the altitude of the air mass above the ground level, and the location of
the experiment together with the altitude at which the air masses are terminated. We have classified
the major air mass origins over the study locations as below:
(a)     A northwest/west (NW/W) sector that includes the North African countries, Arabian

Peninsula, northwestern India (Thar desert), and other Asian countries (such as

Pakistan and Afghanistan) that expect to contribute dust aerosols.

(b)     A northern sector (N), mostly the IGP, with air masses containing highly polluted

aerosols

(c)     Eastern (E) sector with the air masses mostly from the Bay of Bengal of oceanic origin
(d)     Southern (S) sector with the air masses are mostly of oceanic origin;
(e)     Central and Peninsular India (C), where the aerosols are moderately polluted in

comparison with the northern sector

(f)     Local (L) sector in and around the study location where the types of emissions are

heavily dependent on the degree of urbanization.

A similar approach in this way has been made previously by Pawar et al., (2015) over Pune to
obtain the aerosol types from back trajectory analysis. The aerosol types are defined based on Hess
et al. (1998) and brief descriptions of these types are provided in the supplement (**Section S2 and**
**Fig. S3**). The obtained back trajectory clusters and the assigned aerosol types for three locations
are given respectively in **Fig. 5** and **Table 2**. It is also important to note that the scale height of
marine aerosols is typically small (less than 2 km); therefore, we only considered this type within
the boundary layer. Additionally, air masses of oceanic origin that remain on land for more than
24 hours before reaching their destinations are classified as aerosols of continental or local origin.






**Figure 5:** The seven-day air mass back trajectory clusters were analyzed at 500 m and 4000 m above ground level at Gadanki (GDK) (a, b), Hyderabad (HYD) (c, d), and Varanasi (VRN) (e, f).





**Table 2:** Identified aerosol types based on the cluster analysis of air mass back trajectories at the three locations.

| Location | Within Boundary Layer (500m) | Free Troposphere (4 km) |
|---|---|---|
| Gadanki | Maritime Tropical (62%) Continental Average (38%) | Desert (72%) Continental Average (27%) |
| Hyderabad | Urban (59%) Maritime Polluted (41%) | Polluted Continental (44%) Desert (56%) |
| Varanasi | Urban (60%) Continental Average (20%) | Desert (19%) Continental Average (51%) Continental Polluted (30%) |

After obtaining the percentage contribution of each aerosol type in the boundary layer and free
troposphere, we computed the SSA and ASY based on the RH at a given altitude. Suppose $N_i$ is
the fraction of a given aerosol type. In that case, $N_f$ is the total number of aerosol types, and N is
the sum of the fractions of all aerosol types at a given altitude bin (z). SSA and ASY for each
wavelength are obtained according to RH at that particular altitude bin as follows:
$$SSA(z) = \sum_{i=1}^{Nf} \frac{N_i * SSA_i}{N} \qquad (10)$$

$$ASY(z) = \sum_{i=1}^{Nf} \frac{N_i * SSA_i * ASY_i}{N_i * SSA_i} \qquad (11)$$

***4.3. Radiative Transfer Calculations***
For estimating the radiative forcing and heating rates associated with aerosols, we have
incorporated the aerosol data along with the atmospheric parameters and other relevant
information into the Santa Barbara DISORT (discrete ordinate radiative transfer) Atmospheric
Radiative Transfer (SBDART) model (Ricchiazzi et al. 1998). This computational tool calculates
plane-parallel radiative transfer in various atmospheric and surface conditions, including clear and
cloudy scenarios. The DISORT module, which employs a numerically stable algorithm, is used to





solve the equations of plane-parallel radiative transfer in vertically inhomogeneous atmospheres
(Stamnes et al. 1988). The accuracy of the SBDART model is estimated to be within a few percent
for clear-sky conditions (with aerosols), approximately 10% for cloudy-sky predictions of surface
irradiance in the visible spectrum, and possibly as low as 50% for cloudy-sky simulations in the
near-infrared. We performed our calculations in the SW region (0.25–4 μm) with a spectral
resolution of 0.005 μm. The difference between the downward and upward radiative fluxes in
aerosol-laden ( $F_{wa}^{\downarrow}$ and $F_{wa}^{\uparrow}$ respectively) and no-aerosol (and without clouds) ($F_{na}^{\downarrow}$ and $F_{na}^{\uparrow}$ ,
respectively) at the top of the atmosphere (TOA) and the surface of the atmospheric column (SUR)
is referred to as radiative forcing due to aerosols (ARF) at those respective levels. This difference
is mathematically expressed as:
$$ARF_{TOA} = \left(F_{TOA,wa}^{\uparrow} - F_{TOA,wa}^{\downarrow}\right) - \left(F_{TOA,na}^{\uparrow} - F_{TOA,na}^{\downarrow}\right) \qquad (12)$$
$$ARF_{SUR} = \left(F_{SUR,wa}^{\uparrow} - F_{SUR,wa}^{\downarrow}\right) - \left(F_{SUR,na}^{\uparrow} - F_{SUR,na}^{\downarrow}\right) \qquad (13)$$
The atmospheric forcing due to aerosols can be then computed as:
$$ARF_{ATM} = ARF_{TOA} - ARF_{SUR} \qquad (14)$$

Apart from this, we have also calculated the forcing within the boundary layer (from 0 to

2 km) by replacing the TOA to the top of the boundary layer (at 2 km) in equations (12) and (14);
and at the free troposphere (from 2 to 12 km) by replacing TOA to top of the free troposphere (at
12 km) and SUR to bottom of the free troposphere (at 2 km) in (12), (13), and (14). Similarly, the
UTLS forcing is also calculated within the layer from 12 to 20 km.

The ARF calculations are performed using 8 radiation streams at 1-h intervals for a range

of solar zenith angles to obtain a 24-hour average.



The rate at which the atmosphere heats up due to aerosols (referred to as HR, in K day$^{-1}$)
for each layer between the TOA can be determined using the following equation (Liou, 2002):
$$HR = \frac{\partial T}{\partial t} = \frac{g}{C_p}\left(\frac{\Delta F_{ATM}}{\Delta P}\right) = \frac{-1}{\rho C_p}\left(\frac{\Delta F_{ATM}}{\Delta z}\right) \qquad (11)$$

In this equation, 'g' represents the acceleration due to gravity, and '$C_p$' denotes the isobaric
specific heat capacity of dry air ($\sim$ 1006 J Kg$^{-1}$ K$^{-1}$). '$\Delta P$' signifies the pressure difference between
the TOA and SUR boundaries of the atmospheric layer, '$\rho$' indicates the density of the air (in kg
m$^{-3}$), and '$\Delta F_{ATM}/\Delta z$' represents the radiative power absorbed or emitted by the medium per unit
volume of the atmosphere (in Wm$^{-3}$). Since the atmosphere consists of several vertically
heterogeneous layers, repeating the above calculation for each layer yields the profile of the
heating rate profile.
**5. Results and Discussions**
*5.1. Spatial variability of ATAL Aerosols in the UTLS Region*
Fig. 6 shows the mean cloud-screened Backscatter Ratio at 455 nm (BSR455) profiles for
the UTLS region over our study locations.
To approximate the extent of the ATAL region in the UTLS, we utilized the methodology
described by Akhil Raj et al. (2022). According to their approach, the ATAL region's extent is
determined from the convective outflow level - identified by the minimum gradient of potential
temperature below cold point tropopause after smoothing the nine-point running mean- up to the
layer of maximum stability (LmaxS), derived from the square of Brunt- Väisälä frequency (N$^2$).
Their findings indicate that LmaxS is located 1–2.7 km above the cold point tropopause,
corresponding to the potential temperature of approximately 442.11 $\pm$ 25.64 K (454.39 $\pm$ 13.89 K)
over the Indian region, roughly 19 km above the Earth's surface. The convective outflow level,



however, is approximately 13 km across all the locations. Therefore, we defined the approximate
extent of the ATAL region as ranging from 13 km to 19 km 13 km to 19 km (350 K to 440 K
potential temperature).

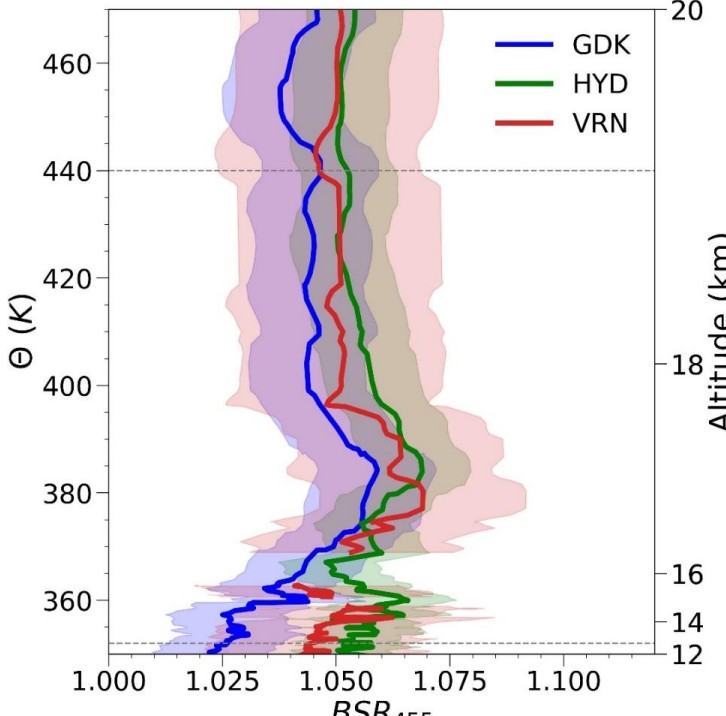


**Figure 6:** The mean cloud-screened backscatter ratios at 455 nm plotted against the potential temperature ($\Theta$) in the primary and altitude in the secondary y-axes over Gadanki (GDK), Hyderabad (HYD), and Varanasi (VRN). The dashed lines at 13 km and 19 km indicate the typical extent of ATAL aerosols.

An increase in aerosols within this altitude range and potential temperature is evident

across all study locations. The $BSR_{455}$ peaks at 1.07 over Varanasi and Hyderabad, followed by
1.06 over Gadanki, with slight variations in the pattern of the backscatter profiles at each location.
These align with the ATAL aerosol patterns observed previously by Akhil Raj et al. (2022) in these
locations. The highest average $BSR_{455}$ observed in this study, 1.07 over Varanasi, is comparable





to observations over Nainital (29.35° N, 79.46° E) in August 2016 (Hanumanthu et al., 2020). The
$BSR_{532}$ of ATAL inferred from CALIPSO was between 1.10 and 1.15 on average, with an
associated depolarization ratio of less than 5% (Vernier et al. 2011). The enhanced $BSR_{455}$ patterns
are more pronounced over Varanasi compared to Hyderabad and Gadanki. Additionally, we
observed greater variability in the backscatter over Varanasi, followed by Hyderabad and Gadanki.
This is consistent with satellite-based ATAL backscatter measurements, which have shown greater
enhancements towards the center of the ASMA region (e.g. Akhil Raj et al., 2022; Vernier et al.,
2015). Supporting these findings, the AODs for the UTLS regions, calculated by integrating the
extinction profiles derived earlier, indicate that the highest mean $AOD_{UTLS}$ (at 500 nm) occurs at
Varanasi and Hyderabad (0.006), followed by Gadanki (0.005). This suggests that the intensity
and complexity of ATAL increase as one moves closer to the center of the ATAL region.
*5.2. Columnar Radiative Forcing Patterns of Sulfate, Nitrate, and Anthropogenic Aerosols in*
*the UTLS:*

The mean columnar radiative forcing at the TOA, surface (SUR), and within the

atmosphere (ATM), estimated for scenarios dominated by sulfate (SUL), nitrate (NIT), and
anthropogenic (ANTH) aerosols in the UTLS region (**Fig. 7**).

The radiative forcing at the TOA ($ARF_{TOA}$) exhibits a negative sign, indicating a net

cooling effect. Among the scenarios, the nitrate-dominated forcing is the most significant,
followed by sulfate and anthropogenic aerosols. This outcome is expected, as the predominance
of scattering aerosols in the UTLS enhances the reflection of incoming solar radiation back to
space, contributing to negative radiative forcing at the TOA.

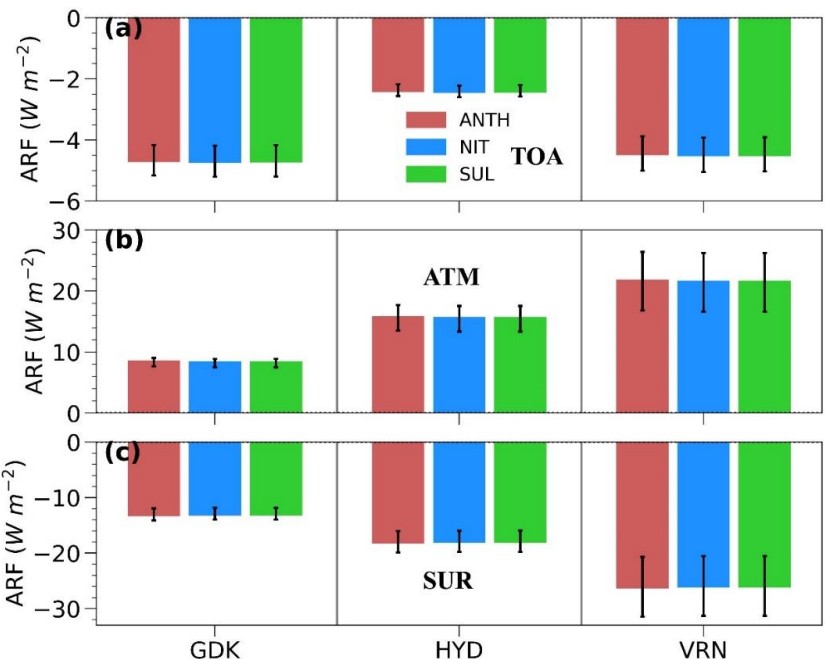

**Figure 7:** Aerosol radiative forcing (ARF) estimates at the (a) top of the atmosphere (TOA), (b) column of the atmosphere (ATM), and (c) surface (SUR) estimated with anthropogenic (ANTH), nitrate (NIT), and sulfate (SUL) dominant scenarios in the UTLS across Gadanki (GDK), Hyderabad (HYD), and Varanasi (VRN).

When analyzing specific locations, we found that $ARF_{TOA}$ estimates are highest over Gadanki, reaching as much as -4.7 ± 0.5 Wm$^{-2}$, followed closely by Varanasi at -4.5 ± 0.6 Wm$^{-2}$, with Hyderabad showing significantly lower values at -2.4 ± 0.2 Wm$^{-2}$. These magnitudes exceed the global average clear-sky aerosol forcing of -1.9 ± 0.3 W m$^{-2}$ reported by Bellouin et al. (2020). Negative aerosol forcing at the TOA has been observed previously across various locations in India under clear-sky conditions. For instance, Santhosh et al. (2024b) reported a forcing -7 ± 0.6 Wm$^{-2}$ at the TOA over Gadanki during the monsoon season, using the long-term CALIPSO aerosol vertical profiles (2006 to 2020). In Hyderabad, previous studies have reported the aerosol radiative forcing ranging from -1 to 7 Wm$^{-2}$ (Gadhavi and Jayaraman, 2006) or as low as -12 Wm$^{-2}$ during



the early monsoon season (Sinha et al., 2013). On a regional scale, the forcing values observed
over Gadanki and Hyderabad in this study are lower than the mean values reported by Kalluri et
al. (2020) over Anantapur (14.62$^{\circ}$ N, 77.65$^{\circ}$ E), where the mean radiative forcing at the TOA was
-6.63 ± 0.77 Wm$^{-2}$. Over Varanasi, our estimates are comparable to those reported by Vaishya et
al. (2018) from the South West Asian Aerosol Monsoon Interactions (SWAAMI) - Regional
Aerosol Warming Experiment (RAWEX) campaign, where they estimated a TOA forcing of -6.5
Wm$^{-2}$ during the onset of the monsoon. However, our estimates are lower than those of Subba et
al. (2022), who found a TOA forcing of -13 ± 1 Wm$^{-2}$ over Varanasi during the monsoon season
using a network of aerosol observatories (ARFINET) combined with concurrent satellite
(CERES)-based TOA fluxes.
The surface forcing across our study locations is also negative. Notably, the anthropogenic
scenario in the UTLS region exhibited the largest magnitudes, followed by nitrate, with sulfate
showing the lowest values. The highest surface forcing was observed over Varanasi at -26 ± 5
Wm$^{-2}$, followed by Hyderabad at -18 ± 2 Wm$^{-2}$ and Gadanki at -13 ± 1 Wm$^{-2}$. The forcing over
Gadanki is consistent with the estimate by Santhosh et al. (2024b) of -16.55 ± 0.64 Wm$^{-2}$, while
the observed forcing over Hyderabad is lower than the estimates found by Sinha et al. (2013),
where they recorded a forcing of approximately -40 Wm$^{-2}$ in August using ground-based
measurements. The surface forcing over Varanasi from our estimates closely matches those of
Subba et al. (2022) at -28 ± 2 Wm$^{-2}$ and Vaishya et al. (2018) at -22.9 Wm$^{-2}$.
Several factors likely contribute to the varying magnitudes of forcing across the locations.
As mentioned earlier, Hyderabad and Varanasi, being heavily urbanized, have a significant
presence of absorbing aerosols within the boundary layer. The highest surface forcing in Varanasi,
followed by Hyderabad, suggests that these absorbing aerosols reduce the amount of solar





radiation reaching the surface by absorbing it, leading to localized cooling with greater intensity compared to the rural background location of Gadanki. Complimenting this observation, we identified atmospheric warming or positive forcing within the atmosphere (ATM), with the highest ATM forcing recorded over Varanasi at $21.62 \pm 4.8$ Wm-2, followed by Hyderabad at $15.6 \pm 2.11$ $Wm^{-2}$ and Gadanki at $8.35 \pm 1$ $Wm^{-2}$. In this case, the anthropogenic aerosol scenario produced the highest forcing, while the sulfate and nitrate estimates were comparable in magnitude. This suggests that the re-emitted thermal energy from the absorbing aerosols is redistributed within the atmospheric column, contributing to warming. The atmospheric forcing over Gadanki and Hyderabad is lower than the estimate of Santhosh et al. (2024b) ($13.85 \pm 0.35$) and Sinha et al. (2013) ($20$ $Wm^{-2}$), respectively. In contrast, our estimates of atmospheric forcing for Varanasi are higher than those reported by Subba et al. (2022) ($15 \pm 1$ $Wm^{-2}$) and Vaishya et al. (2018) ($16.4$ $Wm^{-2}$).

### 5.3. Influence of UTLS aerosols and their composition on the total columnar radiative forcing

To assess the changes in radiative forcing attributable to UTLS aerosols, particularly in terms of their composition, we analyzed the differences in the radiative forcing between scenarios dominated by absorbing aerosols (ANTH) and scattering aerosols (NIT) relative to a sulfate-dominant (SUL) baseline. The difference in aerosol radiative forcing ($\Delta ARF$) was calculated as follows:

$$\Delta ARF_x = ARF_x - ARF_{SUL} \qquad (16)$$

where x represents either the ANTH or NIT scenario. These $\Delta ARF$ values highlight the influence of UTLS aerosols on total columnar radiative forcing.




Our findings indicate that absorption-dominant UTLS aerosols (ANTH) result in positive

radiative forcing at the TOA, while scattering-dominant aerosols (NIT) contribute to a net cooling
effect. This is evident from the positive $\Delta$ARF values for ANTH at the TOA and the negative value
for NIT. The magnitude of these differences is greater in the ANTH scenario than in the NIT
scenario, with $\Delta$ARF for ANTH reaching up to 0.03 Wm$^{-2}$ over Varanasi, followed by 0.02 Wm$^{-2}$
over Hyderabad and Gadanki. In contrast, the $\Delta$ARF for NIT is -0.015 Wm$^{-2}$ across all locations,
indicating that the range of radiative forcing at the TOA varies from -0.015 to 0.03 Wm$^{-2}$ due to
different aerosol scenarios (**Fig.8**).

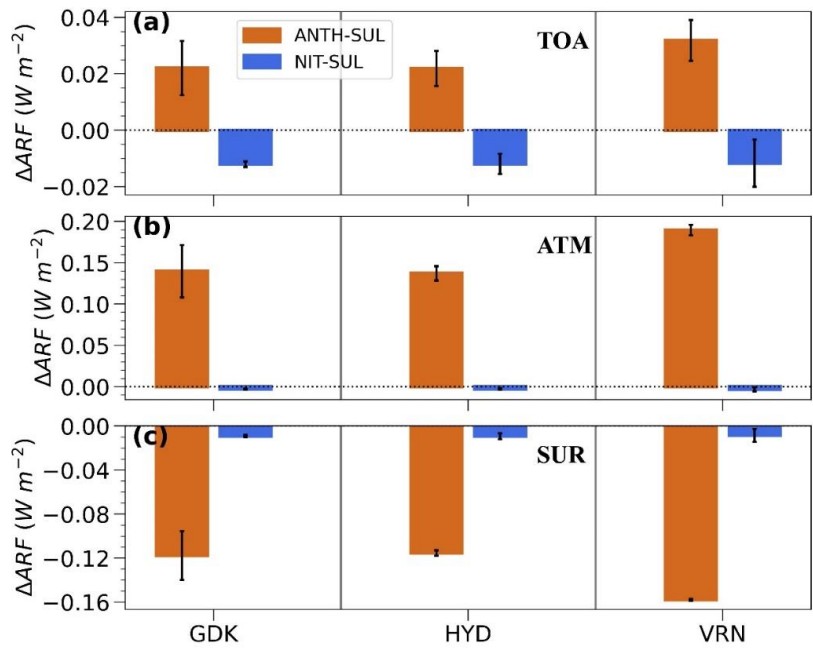


**Figure 8**: The differences in the radiative forcing ($\Delta$ARF) due to the anthropogenic (ANTH) and nitrate (NIT) compositions with respect to sulfate composition at a) Top of the Atmosphere (TOA), (b) Column of the Atmosphere (ATM), and (c) at the surface (SUR)






Our estimates are lower than previous studies, such as Vernier et al. (2015), who reported
a clear-sky radiative forcing of -0.12 Wm$^{-2}$ at the TOA for ATAL aerosols, comparable to the
global radiative forcing from increased $CO_2$ (0.3 Wm$^{-2}$). Gao et al. (2023) reported a positive
forcing of 0.15 Wm$^{-2}$ using the MERRA-2 reanalysis aerosol product. However, they also
observed that the ATAL's impact on TOA forcing varied between -0.002 to 0.15 Wm$^{-2}$ when
comparing different reanalysis and forecast products. These variations underscore the significant
influence of aerosol composition and measurement methodologies on the estimated radiative
forcing. While Vernier et al. (2015) focused on organic carbon and sulfate aerosols, Gao et al.
(2023) included black carbon, which increased atmospheric absorption and, consequently, positive
TOA forcing. In our study, the lower magnitudes, especially in the ANTH scenario, are due to
minimizing the aerosol absorption component (**Sect. 4**).
Interestingly, our findings align more closely with radiative forcing estimates associated
with stratospheric and minor volcanic aerosols. For example, the stratospheric aerosol changes
since 2000 have been estimated at -0.1 Wm$^{-2}$ using near-global satellite aerosol data, offsetting
global warming (Solomon et al., 2011). Similarly, Schmidt et al. (2018) reported a global
multiannual mean forcing of −0.08 Wm$^{-2}$ due to frequent small-to-moderate volcanic eruptions
between 2005 and 2015, relative to the volcanically quiescent period of 1999–2002. Kloss et al.
(2021) documented forcing values ranging from -0.09 ± 0.03 Wm$^{-2}$ to -0.13 ± 0.02 Wm$^{-2}$ due to
the Ulawun eruptions in 2019. Although these studies used aerosol extinction profiles in their
radiative transfer calculations, assumptions about SSA (from 1.0 to 0.97) and ASY (0.5 to 0.85)
were necessary, affecting the results.
In terms of surface radiative forcing ($\Delta ARF_{SUR}$), our analysis shows that both ANTH and
NIT scenarios contribute to surface cooling, with the ANTH scenario having a more substantial



impact. Varanasi exhibited the highest $\Delta ARF_{SUR,\ ANTH}$ (-0.16 $\pm$ 0.001 Wm$^{-2}$), followed by
Hyderabad and Gadanki (-0.12 $\pm$ 0.002 Wm$^{-2}$ and -0.12 $\pm$ 0.02 Wm$^{-2}$, respectively). For $\Delta ARF_{SUR,}$
$_{NIT}$, the impact was consistent across locations (-0.01 Wm$^{-2}$), about one-tenth of the impact from
the ANTH scenario. The overall range of surface forcing due to UTLS aerosols, from -0.01 Wm$^{-2}$
to -0.16 Wm$^{-2}$, though seemingly minor on a local scale, is significant when expressed as -2 Wm$^{-}$
$^{2}$ to -32 Wm$^{-2}$ per unit AOD (at 500 nm). This implies that even small increases in UTLS aerosol
loading can substantially enhance surface cooling over time.
In the atmospheric column ($\Delta ARF_{ATM}$), ANTH aerosols were found to enhance
atmospheric warming, whereas NIT aerosols contributed to atmospheric cooling. As with TOA
forcing, the differences were more pronounced in the ANTH scenario. The highest $\Delta ARF_{ATM,\ ANTH}$
was observed over Varanasi (0.19 $\pm$ 0.001 Wm$^{-2}$), followed by Hyderabad and Gadanki (0.14 $\pm$
0.01 Wm$^{-2}$ and 0.14 $\pm$ 0.03 Wm$^{-2}$, respectively). Interestingly, $\Delta ARF_{ATM,\ NIT}$ was negligible (-0.003
Wm$^{-2}$) across all locations, indicating that the influence of NIT-dominant UTLS aerosols in the
atmospheric column is nearly indistinguishable from that of the background sulfate aerosols. This
negligible difference can be attributed to the similar SSA values associated with nitrates and
sulfates, as noted by Zhang et al. (2012). In this study, the SSA of the UTLS region remained at 1
across both wavelengths (455 nm and 940 nm) (**Fig. 4**), suggesting that any differences in
atmospheric forcing estimates between sulfates and nitrates are likely due to slight variations in
the column ASY of the UTLS region.  In the case of Gadanki, the column asymmetry parameter
(ASY) for the UTLS at 455 nm (940 nm) with sulfate aerosols (SUL) is recorded at 0.76 (0.72),
whereas for nitrate aerosols (NIT), it is 0.74 (0.70). This has led to the application of sulfate aerosol
properties in the estimation of the radiative impacts of nitrates, a method previously employed by
various researchers. However, it is important to note that there are significant differences in the



single scattering albedo (SSA) between nitrates and sulfates at specific wavelengths. For instance,
at a wavelength of approximately 2.8 μm and relative humidity (RH) below 40%, the SSA for
nitrates is about 40% higher than that for sulfates (Zhang et al., 2012). This indicates that nitrate
aerosols could be more absorptive at these wavelengths, leading to considerable radiative impacts
in the UTLS nitrate aerosol column. Since we lack direct measurements at these wavelengths, the
radiative impacts of sulfate and nitrate aerosols in the atmospheric column may appear similar.
However, this similarity should not be interpreted as a justification for substituting the optical and
microphysical properties of nitrate aerosols with those of background sulfate aerosols.
*5.4. Impact of Aerosol Radiative Forcing across Atmospheric Layers*
The contribution of aerosol radiative forcing within different atmospheric layers - the
boundary layer (0 to 2 km), the free troposphere (2 to 12 km), and UTLS (12 to 20 km) - to the
total columnar aerosol radiative forcing was evaluated for three different aerosol compositions
(**Fig. 9**).
Our analysis revealed that the ANTH dominant scenario exhibited the highest radiative
forcing values within the UTLS, with the greatest magnitude over Varanasi ($0.25 \pm 0.09$ Wm$^{-2}$),
followed by Hyderabad ($0.22 \pm 0.02$ Wm$^{-2}$) and Gadanki ($0.2 \pm 0.08$ Wm$^{-2}$). In contrast, the NIT
scenario showed much lower forcing values in the UTLS, with $0.02$ Wm$^{-2}$ over Gadanki and
Hyderabad and a slightly higher value of $0.03$ Wm$^{-2}$ at Varanasi. This suggests that the presence
of absorbing aerosols in the UTLS leads to localized warming, whereas scattering aerosols
contribute to minimal or negligible warming. In terms of percentage contribution to the total
columnar atmospheric forcing, the UTLS contributes between 0.1% and 2.3% across all locations.
Consistent with previous findings, the ANTH dominant scenario in the UTLS contributes the most




(1.4% to 2.3%), while the contributions from the NIT and SUL scenarios are significantly lower
(0.1% to 0.2%).

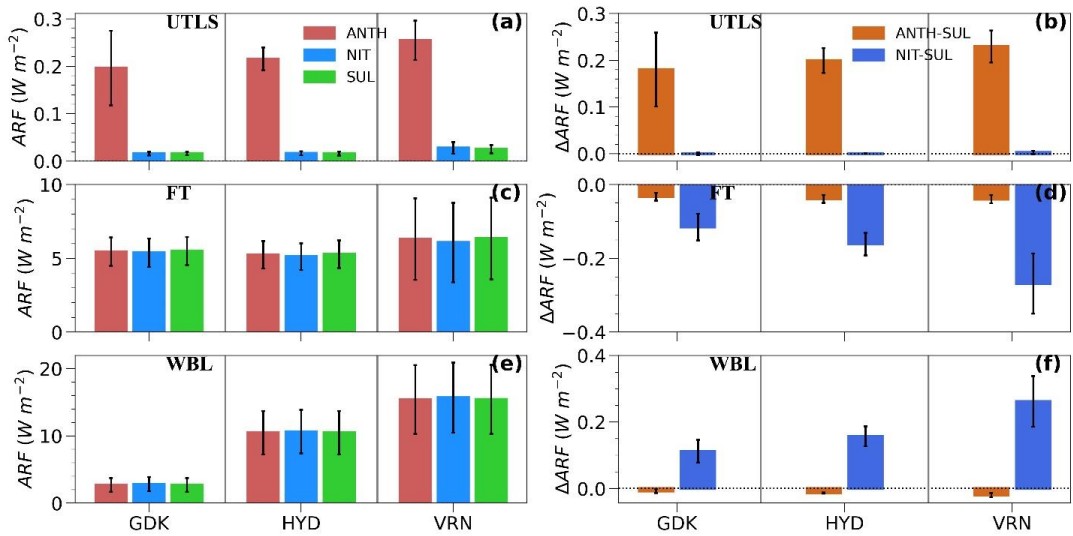

**Figure 9:** The contribution of (a) UTLS, (b) Free Troposphere (FT), and (c) Within the boundary layer (WBL) forcing towards the total columnar atmospheric forcing. The right panel (b, d, and f) shows the ΔARF due to ANTH and NIT scenarios with respect to the SUL at UTLS, free troposphere, and within the boundary layer, respectively.

The right panels of **Fig. 9 (b, d, f)** illustrate the changes in radiative forcing within the
boundary layer, free troposphere, and UTLS when transitioning from SUL to either the ANTH or
NIT scenarios (ΔARF). Overall, ΔARF values were highest over Varanasi, followed by Hyderabad
and Gadanki. Under the ANTH conditions, a slight decrease in radiative forcing was observed in
the boundary layer and free troposphere (up to -0.02 Wm$^{-2}$ in the boundary layer and -0.04 Wm$^{-2}$
in the free troposphere). In the UTLS, the transition from SUL to ANTH resulted in a significant
increase in radiative forcing, whereas the change from SUL to NIT was minor or negligible. These
observations suggest that ANTH aerosols in the UTLS absorb incoming solar radiation, leading to





localized heating in that layer. This absorption reduces the amount of solar radiation reaching the
free troposphere and boundary layer.
The complex interactions between scattering and absorbing aerosols in the free troposphere
further reduce the amount of radiation reaching the boundary layer, eventually leading to decreased
radiative forcing within the boundary layer. The $\Delta ARF_{ANTH}$ per unit AOD of the UTLS (at 500
nm) ranged from -2 to -4 $Wm^{-2}$ in the boundary layer and free troposphere. These changes in the
anthropogenic aerosol loading in the UTLS could have a non-negligible long-term impact on the
dynamics of the boundary layer and free tropospheric aerosols.
The similarity in radiative forcing estimates between the SUL and NIT scenarios in the
UTLS can be attributed to their similar scattering properties, as discussed earlier. However,
$\Delta ARF_{NIT}$ was found to be positive in the boundary layer and negative in the free troposphere. This
is intriguing because it suggests that scattering-dominant aerosols in the UTLS slightly warm the
boundary layer while cooling the free troposphere. The scattering effect of nitrates in the UTLS
might redistribute solar energy, causing more energy to be scattered and absorbed in the free
troposphere, thereby increasing local radiative forcing. In comparison, the boundary layer receives
less energy, leading to a reduction in forcing.
Another possibility is that increased scattering in the UTLS counteracts warming in that
layer, stabilizing the atmosphere and reducing vertical mixing. This stabilization could isolate the
free troposphere, allowing it to retain more of the scattered energy from the lower atmosphere,
thereby increasing local radiative forcing. Consequently, the reduced radiant energy reaching the
boundary layer leads to the observed cooling effect.





It is also important to note that apart from the ATAL, numerous studies have reported the
presence of elevated aerosol layers (EALs) in the free troposphere during the pre-monsoon and
monsoon months over the Indian region due to long-range transport and vertical convective lofting
of aerosols (Kumar et al., 2023; Gupta et al., 2021; Niranjan et al., 2007; Ratnam et al., 2018;
Sarangi et al., 2016). These EALs lead to significant radiative impacts, including lower
tropospheric cooling due to increased aerosol absorption and scattering, which affects the regional
climate and atmospheric stability. Such impacts due to EALs during monsoon months also
contribute to the varied influence of the UTLS aerosols in the boundary layer and free troposphere.
*5.5. Aerosol Heating Rates and their Implications in the UTLS Region*
Consistent with earlier observations of radiative forcing in this study, heating rates
associated with the ANTH composition are significantly higher than those observed in the SUL
and NIT scenarios (**Fig. 10**). Specifically, the heating rates in the SUL and NIT scenarios are nearly
10 times smaller than those in the ANTH dominant scenario. Neither the SUL nor NIT scenarios
displayed distinct heating rate patterns at the ATAL altitudes, with only slight warming observed,
reaching a maximum of 0.003 K day$^{-1}$. This lack of difference in heating rate patterns between
sulfate and nitrate aerosols can be attributed to their similar optical properties, as discussed in
previous sections.
In contrast, under the ANTH composition, enhanced heating rates were observed from 16
km to 18 km over Gadanki (365 to 404 K potential temperature), from 14km to 16 km (357 to 366
K potential temperature) over Hyderabad, and from 16 km to 18 km (367 to 403 K potential
temperature) over Varanasi. This heating in the ATAL layer reached as high as 0.03 K day$^{-1}$,
indicating a slight warming due to the presence of absorbing aerosols in the UTLS region. The
heating rates over Varanasi were notably higher than those at the other locations. Previous studies

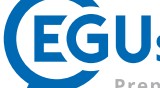

support these findings; for instance, Fadnavis et al. (2022) reported UTLS warming due to
anthropogenic aerosols, with estimates ranging from 0.02 to 0.3 K per month. Similarly,
carbonaceous aerosols, which have strong absorption characteristics, increased UTLS heating by
0.001 to 0.02 K day$^{-1}$ (Chavan et al., 2021).

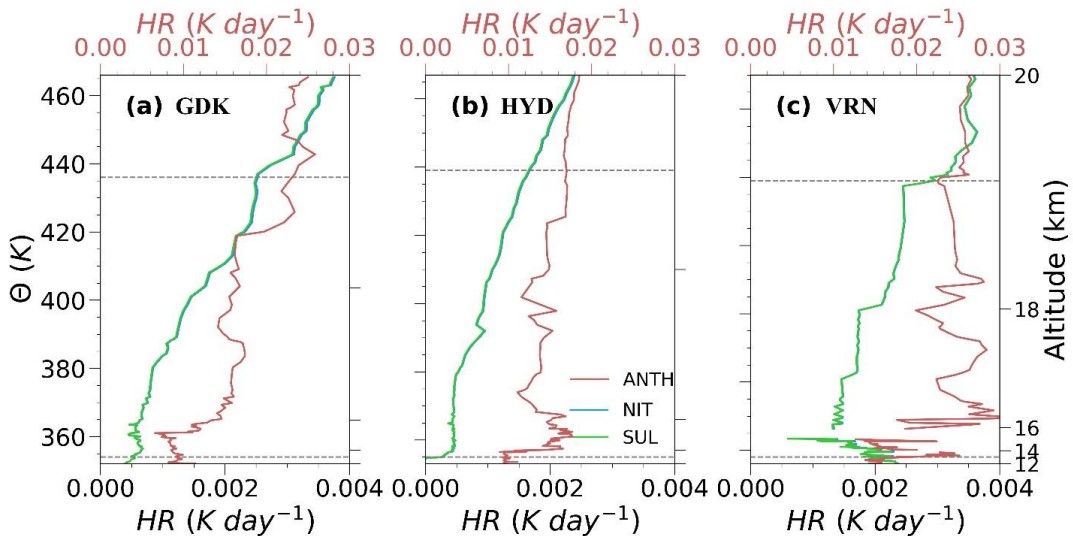


**Figure 10:** The aerosol heating rates at the UTLS plotted against potential temperature (Θ) and altitude over (a) Gadanki, (b) Hyderabad, and (c) Varanasi. The grey dashed lines at 13 km and 19 km represent the approximate extent of ATAL.

This warming in the UTLS region could have several significant consequences. As
warming occurs in the boundary layer and free troposphere, the elevated temperature in the UTLS
could lead to an increase in water vapor concentration in the lower stratosphere. Fadnavis et al.
(2022) noted that South Asian aerosols contribute to enhanced water vapor levels in the lower
stratosphere at tropical and subtropical latitudes. As a potent greenhouse gas, increased water
vapor in the UTLS region could amplify warming through positive feedback mechanisms. Huang
et al. (2016) estimated a weak positive global-mean radiative feedback ($0.02 \pm 0.01$ W m$^{-2}$ K$^{-1}$)



due to increased stratospheric water vapor concentration. Furthermore, Solomon et al. (2010)
estimated that a 1 ppmv increase in water vapor could lead to a global average radiative forcing of
0.24 W m$^{-2}$ at the TOA, comparable to the 0.36 W m$^{-2}$ increase in radiative forcing due to the
growth of carbon dioxide from 1980 to 1996.

Another potential consequence of increased water vapor in the lower stratosphere is ozone

depletion. For example, box-model simulations by Robrecht et al. (2019) showed that high water
vapor mixing ratios could lead to approximately 20% of ozone destruction through catalytic ozone
loss cycles. In addition, UTLS warming and associated increases in water vapor could influence
aerosol microphysical properties. Balloon-borne measurements by He et al. (2019) revealed that
larger particles in the UTLS aerosol layer, which are generally very hydrophilic, experience
dramatic size increases with rising relative humidity. These size changes can further alter the
scattering and absorption characteristics of aerosols, leading to varied radiative impacts.
Consequently, the dominant presence of absorbing aerosols in the UTLS has the potential to create
a complex feedback mechanism, influencing the radiative balance by altering the compositions of
water vapor, ozone, aerosols, and trace gases.
**6. Summary and Conclusions**

This study provides a detailed analysis of the radiative impacts of monsoon UTLS aerosols,

focusing on radiative forcing and heating rates, based on balloon-borne in situ measurements from
the BATAL field campaigns conducted between 2014 and 2018. To assess the aerosol effects,
three idealized scenarios were considered, each dominated by a different type of aerosol: (i) Sulfate
(SUL) representing the background or reference condition, (ii) Nitrate (NIT) for scattering-
dominant aerosols, and (iii) Anthropogenic (ANTH) for absorption-dominant aerosols. The key

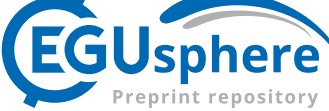



findings from the study, conducted over three locations - Gadanki, Hyderabad, and Varanasi - are
summarized below:
i)    **Aerosol Enhancement in ATAL Altitudes**: A significant increase in aerosol
concentrations was observed at the ATAL altitudes (13 to 19 km) across all locations, with
$BSR_{455}$ peaks reaching as high as 1.07 over Varanasi and Hyderabad, followed by 1.06
over Gadanki. The highest mean AOD in the UTLS was recorded over Varanasi, followed
by Hyderabad and Gadanki, indicating the strength and complexity of ATAL vary from
the edge to the center of the ASMA region.
ii)   **Radiative Forcing across Layers**: The study found cooling effects at the top of the
atmosphere (TOA) ranging from $-2.37 \pm 0.19$ Wm$^{-2}$ to $-4.5 \pm 0.6$ Wm$^{-2}$ and at the surface
from $-12.9 \pm 1$ Wm$^{-2}$ to $-26 \pm 5$ Wm$^{-2}$. At the same time, warming was observed within the
atmospheric column, ranging from $8.21 \pm 0.68$ Wm$^{-2}$ to $21.62 \pm 4.8$ Wm$^{-2}$. The ANTH
scenario showed the highest radiative forcing magnitudes in the UTLS, nearly comparable
with the SUL and NIT scenarios.
iii)  **Changes in Radiative Forcing (ΔARF)**: The influence of UTLS aerosols on radiative
forcing at TOA, surface (SUR), and within the atmosphere (ATM) was quantified by
comparing the changes from the reference SUL composition to the NIT and ANTH
compositions. The ΔARF due to ANTH aerosols was positive (indicating warming) at TOA
and ATM but negative (indicating cooling) at the surface. Conversely, ΔARF due to NIT
aerosols was negative across TOA, SUR, and ATM. The radiative forcing changes in the
TOA ranged from $-0.015$ to $0.03$ Wm$^{-2}$, at the surface from $-0.01$ Wm$^{-2}$ to $-0.16$ Wm$^{-2}$, and
within the atmosphere from 0 to 0.19 Wm$^{-2}$. These changes are similar to the radiative
impacts of minor volcanic eruptions and could have long-term effects on regional weather





patterns. The most significant impacts were observed over Varanasi, followed by
Hyderabad and Gadanki.
iv)  **Contribution of UTLS Aerosols to Total Columnar Forcing**: The UTLS aerosols
contributed between 0.1% and 2.3% of the total columnar atmospheric forcing. The ANTH
scenario had the highest contribution (1.4% to 2.3%), while the NIT and SUL scenarios
contributed significantly less (0.1% to 0.2%). The highest forcing estimates in the UTLS
column were over Varanasi ($0.25 \pm 0.09$ Wm$^{-2}$), followed by Hyderabad ($0.22 \pm 0.02$ Wm$^{-2}$
) and Gadanki ($0.2 \pm 0.08$ Wm$^{-2}$) under the ANTH scenario. Under the NIT scenario, the
forcing values were 0.02 Wm$^{-2}$ over Gadanki and Hyderabad, with a slightly higher value
of 0.03 Wm$^{-2}$ over Varanasi.
v)  **Impact on Boundary Layer and Free Troposphere**: UTLS aerosols also influenced the
radiative balance within the boundary layer (WBL) and free troposphere (FT). Under the
ANTH scenario, a slight decrease in radiative forcing (cooling) was observed in the WBL
and FT (up to -0.04 Wm$^{-2}$). In contrast, the NIT scenario resulted in a slight increase in the
WBL (up to 0.26 Wm$^{-2}$) and a slight decrease (cooling) in the FT (up to -0.27 Wm$^{-2}$).
vi)  **Heating Rate Profiles in UTLS**: The heating rate profiles for the UTLS under the ANTH
scenario showed a marked increase in aerosol heating at the ATAL altitudes, with the
highest rates recorded over Varanasi (up to 0.03 Kday$^{-1}$), compared to other locations.
However, the heating rates under the SUL and NIT scenarios were nearly one-tenth of
those under the ANTH scenario, indicating significantly lower heating.
Overall, this study demonstrated that ATAL aerosols in the UTLS have diverse impacts on
different atmospheric layers, varying across geographic locations within the ATAL region. The
scattering and absorption properties of the aerosols present strongly influence these impacts. The



actual composition of ATAL is likely complex, given the chemical and dynamic variability within
the region (e.g., Hanumanthu et al., 2020). Therefore, altitude-resolved aerosol composition data
from real-time measurements are crucial for accurately assessing their radiative impacts, which
also applies to aerosols within the boundary layer and free troposphere. It is important to emphasize
that the radiative forcing and heating rates of nitrate and sulfate aerosols are comparable within
the atmospheric column and UTLS due to their similar scattering properties across a broad range
of shortwave spectrum wavelengths. However, to capture the specific absorption characteristics of
nitrate aerosols at particular wavelengths, improved measurement techniques in the thermal
infrared and longwave regions are needed. Enhancing these measurements will also aid studies on
UTLS water vapor, a potent greenhouse gas whose radiative impacts are more sensitive in the
longwave region (e.g., Santhosh et al., 2024a).
The strength of ATAL backscatter ratios varies geographically within the ASMA region,
leading to corresponding variations in radiative forcing and heating rates. The highest values were
observed over Varanasi, a more centrally located area within the region. In contrast, estimates for
Hyderabad and Gadanki, located near the edges of the ATAL region, were comparable. However,
further verification of this pattern on a global scale across the entire ATAL region is necessary.
Given the geographic limitations of in-situ measurements, combining satellite data, reanalysis, and
in-situ measurements is essential, and efforts in this direction are currently underway. Moreover,
a series of experiments as part of the second phase of the BATAL campaigns, concluded in August
2024, are expected to provide new insights into ATAL research, particularly regarding the
influence of wildfires and volcanic eruptions.




### Code/Data availability

The data collected from the BATAL campaigns is available on request.

### Author contribution

**V.N. Santhosh:** Data curation, Formal analysis, Investigation, Software, Validation, Visualization, Writing – original draft. **B.L. Madhavan:** Conceptualization, Investigation, Methodology, Supervision, Writing – review & editing. **S.T. Akhil Raj:** Data curation, Visualization, Writing – review & editing. **M. Venkat Ratnam:** Project administration, Resources, Writing – review & editing. **J-P. Vernier:** Project administration, Resources, Writing – review & editing. **F.G. Wienhold:** Software, Writing – review & editing.

### Competing interests

The authors declare that they have no conflict of interest.

### Acknowledgements

The findings presented in this paper are derived from the ISRO-NASA joint BATAL campaign, which was supported by the National Atmospheric Research Laboratory (NARL) under the Department of Space (DoS), and NASA ROSES Upper Atmospheric Research Program and Atmospheric Composition Modeling and Analysis Program (UARP, ACMAP, UACO). We extend our gratitude and acknowledge Dr Amit Kumar Pandit, National Institute of Aerospace, Hampton, USA, and other members from NARL Gadanki, TIFR Balloon Facility Hyderabad, and BHU Varanasi for their active involvement in the BATAL campaigns from Gadanki to Varanasi. We thank NASA's Earthdata team for providing free access to their MERRA-2, MODIS, MLS, and AIRS datasets which were used as supportive data in this study. We also thank the National Oceanic and Atmospheric Administration (NOAA)'s Air Resources Laboratory (ARL) for their HYSPLIT software.

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
