# Peer review of "Shortwave Radiative Impacts of the Asian Tropopause Aerosol Layer (ATAL)"

_EGUsphere, 2024_

## Author Comment (AC1)

**Replies to Reviewer # 2 Comments/Suggestions**

**General Comments:**

The paper provides a comprehensive review of the key findings related to the Asian Tropopause Aerosol Layer (ATAL). It utilizes data from balloon measurements conducted at three different locations in India to construct mean profiles of aerosol backscatter, water vapor, ozone, and other parameters for the UTLS region. Additionally, the authors present and discuss the computed radiative forcings and heating rates based on three different aerosol scenarios for ATAL. Overall, the paper serves as a valuable review of the ATAL observed over the Indian region, although it is important to note that the full extent of ATAL is much broader.

**Reply:** *We sincerely appreciate the reviewer's thorough summary of our study. Based on the insightful comments and suggestions provided, we have made significant revisions to improve the manuscript.*

**Specific Comments:**

The major limitation of this study is the lack of detailed information on the size-resolved chemical composition of the ATAL. In the absence of this data, the authors propose three scenarios—sulfates, nitrates, and anthropogenic particles—to represent the composition of ATAL and calculate the corresponding radiative forcing and heating rates. Previous studies have investigated the radiative forcing of ATAL aerosols, focusing on components such as organic carbon, black carbon, and sulfates. However, in reality, ATAL is a mixture of all these aerosol types, and its composition varies from the outer to the central regions. Therefore, characterizing ATAL simply as consisting of nitrates, sulfates, or anthropogenic particles, as done in the present study, oversimplifies the actual complexity of its composition and the obtained radiative forcing and heating rates can be far from reality.

**Reply:** *We acknowledge the reviewer's concern regarding the simplified representation of ATAL composition in our initial analysis. In the revised manuscript, we now account for ATAL as a mixture of aerosol species—sulfates, nitrates, organic carbon, and ammonia—by incorporating composition estimates from recent modeling and observational studies. This approach better reflects the spatial variability and complexity of ATAL's composition, providing a more representative assessment of its radiative effects.*

**Technical corrections:**

L46: The geographic area of the ATAL is described as 10 to 40 deg N and 10 to 140 deg E in page 2 line 46 and in Fig 1 caption it is shown as 15 to 40 deg N and 15 to 105 deg E, whereas Gadanki is at 13.5 deg N which is however shown within the red box in the figure.

**Reply:** *We thank the reviewer for bringing this to our attention. We have corrected the figure in the revised manuscript.*

L131: it is mentioned that "These campaigns involved over a hundred balloon flights.." whereas Table 1 shows a total of 21 flights.

**Reply:** *We appreciate the reviewer's observation. The campaign indeed included around a hundred balloon flights; however, for this study, we specifically selected only those that reached at least 20 km altitude to ensure continuous backscatter and atmospheric profiling. Many flights did not meet this criterion, as they were intended for other objectives, allowing only a few to float within the 16 km to 18 km range. Additionally, some flights lacked ozonesondes/COBALD sensors. Consequently, 21 flights were selected for our analysis. To avoid any ambiguity, we have revised this statement in the manuscript for better clarity.*

Fig 4:  ASY for nitrate shows a dip at 100% RH which is difficult to comprehend.

**Reply:** *We thank the reviewer for bringing this to our attention. The dip in ASY at 100% RH was due to a technical plotting error, which has now been corrected. In the revised manuscript, we have refined the size distribution parameters for nitrate aerosols—mode radius (0.15 μm) and standard deviation (2 μm) — within the 0.5–2 μm size range, following Zhang et al. (2012) and Vernier et al. (2022). Additionally, we have revised the SSA and ASY calculations across different RH bins using the methodology from Zhang et al. (2012). Since the UTLS remains predominantly dry, we now present SSA and ASY values for nitrates under dry conditions (zero RH) across altitude bins in the updated manuscript.*

[Figure]

*Figure R1: Single Scattering Albedo (SSA) and Asymmetry Parameter (ASY) for nitrate aerosols at selected Relative Humidity (RH) levels. In the revised analysis, SSA and ASY values for nitrates are considered under dry conditions (zero RH) due to the predominantly dry nature of the UTLS.*

L181-184: It is mentioned that the measured water vapour values over Varanasi are higher than that over Hyderabad and Gadanki. However, it is difficult to appreciate this fact in Figure 2c, unless log scale is used for the concentration values.

**Reply:** *We appreciate the reviewer's suggestion. In the revised manuscript, we have updated Figure R2 to use a logarithmic scale for water vapor concentration, making the differences*

*between locations more apparent. To maintain the manuscript's focus on the radiative impacts of UTLS aerosols, we have moved this figure to the supplementary section.*

[Figure]

*Figure R2: Vertical profiles of (a) Pressure, (b) Temperature, (c) Water Vapor Density (WV; log scale), and (d) Ozone Density (O₃) in the UTLS region for Gadanki (GDK), Hyderabad (HYD), and Varanasi (VRN). The log scale in panel (c) enhances the visibility of variations in water vapor concentration across locations.*

Table 2: Aerosol types within boundary layer for Varanasi, Urban (60%) and Continental Average (20%) doesn't add to 100%.

**Reply:** *We appreciate the reviewer for bringing this to our attention. This was a typographical error— 'Urban' aerosols contribute 80%, not 60%. We have corrected this in the revised manuscript. Additionally, since the primary focus of the study is on UTLS aerosols, we have moved this table to the supplementary information.*

L431: "13 km to 19 km" repeated twice.

**Reply:** *We have corrected this in the revised manuscript.*

Equ 16: Since DARFx is the difference between relatively two large numbers it can be within the uncertainty of the computed ARF.

**Reply:** *We agree with the reviewer's observation. To improve clarity, we have revised the notation from $\Delta ARFx$ to $\delta ARFx$, distinguishing it from the total forcing of a given layer. The computed $\delta ARFx$ represents atmospheric forcing differences from the surface (0 km) to the top of the atmosphere (30 km). While $\delta ARFx$ values reach up to 0.5 W m$^{-2}$, they remain smaller than the standard deviation of the total column ARF (up to 5 W m$^{-2}$), indicating that UTLS aerosol composition changes have a minor impact on total atmospheric forcing. However, within the UTLS region itself, $\delta ARFx$ values range from 0.06 to 0.28 W m$^{-2}$, exceeding the standard deviations of UTLS forcing (0.01–0.08 W m$^{-2}$). This suggests that $\delta ARFx$ in the UTLS is significant, highlighting localized but strong radiative effects.*

*We sincerely thank the reviewer once again for valuable suggestions, which have significantly enhanced the content of the manuscript.*

---

## Author Comment (AC2)

**Replies to Reviewer # 1 Comments**

**General Comments:**

The goal of this study is to evaluate the direct radiative impacts of the Asian Tropopause Aerosol Layer (ATAL) aerosols in the upper troposphere and lower stratosphere (UTLS) using in-situ measurements from the Balloon measurement of the Asian Tropopause Aerosol Layer (BATAL) campaigns (2014-2019). Detailed analyses of various aspects of the aerosol radiative forcings are presented using extensive sources of measurements and methods. I think this work will have significant contributions to the wide scientific community. I have four major comments in addition to various technical comments for the authors might take into consideration.

**Reply***: We sincerely appreciate the reviewer's positive assessment of our study and its relevance to the broader scientific community. In response to the reviewer's insightful comments and suggestions, we have carefully revised the manuscript to improve clarity, enhance the analysis, and address the key concerns raised. We believe these revisions have strengthened the overall presentation and scientific rigor of the study.*

Extensive analyses with various measurements and complicated methodology are presented in this work. The goal of this study should be mentioned clearly along with what the new findings are and why this work is important in the scientific community. In many places, the results are compared with previous studies without emphasizing the unique value of the results presented in this work.

**Reply:** *We appreciate the reviewer's feedback on clarifying the study's objectives and emphasizing its unique contributions. In response, we have revised the introduction to define the study's goals and highlight its significance clearly. The key objectives are:*

1. *Characterizing ATAL enhancements across multiple BATAL study locations (Gadanki, Hyderabad, and Varanasi) during the monsoon season.*
2. *Quantifying ATAL radiative forcing and heating rates within the UTLS (12–20 km) using high-resolution in-situ measurements.*
3. *Assessing the contribution of ATAL to total atmospheric column forcing, providing insights into their relative radiative impact.*
4. *Evaluating the sensitivity of ATAL radiative forcing and heating rates to variations in aerosol compositions.*
5. *Comparing the heating rate estimates from in-situ measurement at three-point locations with spatially averaged estimates derived from satellite observations to assess regional representativeness and variability.*

*To achieve these objectives, we introduce a methodological framework that integrates in-situ data for cloud screening, aerosol type identification, and radiative transfer calculations. The revised manuscript presents these objectives in the introduction and refines comparisons with previous studies to highlight the novel aspects of our findings better.*

This study considers three scenarios where UTLS aerosols are predominantly composed of sulfates, nitrates, or anthropogenic aerosols. I am not sure if this is a new approach or has been used in the past. There are detailed explanations about this approach throughout the manuscript, but it is unclear if this method is new to this work or not.

**Reply:** *We appreciate the reviewer's query regarding the novelty of our approach. The consideration of three aerosol scenarios—sulfates, nitrates, and anthropogenic aerosols—was initially based on our recent work (Santhosh et al., 2025), which quantified the radiative forcing and heating rates of ATAL over a broader ASMA region using long-term satellite backscatter measurements. However, the methodology in that study differs from the present work, as it focuses on satellite retrievals, whereas this study is based on in-situ balloon-borne COBALD measurements.*

*To address potential oversimplifications in ATAL composition, we have expanded the analysis in the revised manuscript to include seven aerosol composition scenarios (Table 3). These include:*

- *Five scenarios representing external mixtures of sulfates, nitrates, organic carbon, and ammonium aerosols,*
- *Two scenarios dominated by either sulfates or nitrates, similar to our original assumption.*

*This refined approach integrates compositions reported by Bossolasco et al. (2021) and Appel et al. (2022) while ensuring that aerosol fractions remain within the range of observed values. The revised methodology enhances the alignment with real-world ATAL conditions and improves the robustness of our radiative forcing estimates.*

This work needs more focus. In addition to in-situ measurements, sources including reanalysis (MERRA-2) and a model (HYSPLIT) are also used. It is useful to include many data sources and results from them, but it makes it somewhat harder to focus on the main point of this study.

**Reply:** *We appreciate the reviewer's feedback on maintaining a clear focus in this study. The inclusion of non-in-situ data sources, such as MERRA-2 reanalysis and HYSPLIT trajectory modeling, is intended to provide complementary context rather than being the primary focus. These datasets support our interpretation of ATAL aerosol transport, composition, and variability. To enhance clarity, we have streamlined the methodology section in the revised manuscript, ensuring that the core analysis remains centered on in-situ measurements. Additional details on MERRA-2 and HYSPLIT have been moved to the supplement, allowing readers to access them without diverting from the study's primary objectives.*

Some part of the manuscript includes general statement about the impacts on climate without solid evidence. Reducing those statements would make this study concise and therefore easier to read.

**Reply:** *We appreciate the reviewer's suggestion to enhance conciseness and clarity. In response, we have carefully reviewed the manuscript and removed or refined general*

statements about climate impacts that lacked direct supporting evidence. The results and discussion sections have been revised to ensure they remain focused and align closely with the well-defined objectives of this study.

**Specific Comments:**

L14 (Abstract) - The abstract does not include all the important findings in this work. It needs to emphasize what the new results are without explaining the background information in abstract.

**Reply:** *We appreciate the reviewer's feedback on improving the abstract. In response, we have restructured the abstract in the revised manuscript to focus on key findings and highlight the novel aspects of this study. Background information has been reduced to ensure a clear and concise presentation of the most significant results.*

L42 – Needs a reference (references) at the end of the sentence.

**Reply:** *Included in the revised manuscript.*

L47 - Add more recent references on the Asian monsoon pollution, *e.g.*, von Hobe et al. (2021, ACP).

**Reply:** *Included in the revised manuscript.*

L52- Add more specific information about how the ATAL impacts earth's radiative balance based on previous studies, as it is the focus of this paper.

**Reply:** *We appreciate the reviewer's suggestion. In the revised manuscript, we have expanded the discussion on how ATAL influences Earth's radiative balance, incorporating more specific details from previous studies. Additional references have been included to provide a stronger context for this study's focus.*

L58 – What do 'simulations' mean? Please specify (*e.g.*, global, regional, or trajectory model simulations).

**Reply:** *We appreciate the reviewer's clarification request. In the revised manuscript, we have streamlined the discussion on ATAL transport pathways into a concise statement, rather than listing individual results from each reference. As a result, the original statement, which referred to coupled aerosol-chemistry-climate model simulations by Fadnavis et al. (2013), has been removed for improved clarity.*

L72 – A brief explanation about what the secondary aerosol formation would be helpful here.

**Reply:** *We appreciate the reviewer's suggestion. In the revised manuscript, we have clarified that secondary aerosol formation refers to the gas-to-particle conversion of inorganic and organic precursors, as inferred from the chemical analysis by Appel et al. (2022).*

L80 – What does 'Influence the extinction' mean? Did it increase or decrease the extinction?

**Reply:** *We appreciate the reviewer's clarification request. Ma et al. (2019) reported an increase in aerosol extinction in the upper troposphere over Tibet, mainly due to mineral dust aerosols. To improve clarity and streamline the introduction, we have removed the original statement but retained the citation, as the study provides valuable insights into ATAL aerosol composition, which is now referenced in Table 2 of the revised manuscript.*

L93- What is 'STP'?

**Reply:** *STP stands for Standard Temperature and Pressure (298 K and 1013.25 hPa). To improve the conciseness of the introduction, this sentence has been removed, while the study is still cited in Table 2 of the revised manuscript, as it provides valuable information about the ATAL composition.*

L100-101 – It would be useful to explain what the complexity of retrieving the aerosol properties are here.

**Reply:** *We appreciate the reviewer's suggestion. Retrieving aerosol properties in the UTLS is challenging due to low aerosol concentrations, instrumental detection limits, and uncertainties in satellite and lidar retrievals. Additionally, complex transport and mixing processes make it challenging to determine the chemical composition of aerosols, which is crucial for deriving scattering properties, such as the single scattering albedo (SSA) and asymmetry parameter (ASY), for radiative transfer calculations. These complexities are now explicitly addressed in the revised manuscript.*

L104 – Is the negative radiative forcing due to scattering by the aerosols?

**Reply:** *Yes. The negative radiative forcing reported by Vernier et al. (2015) is primarily attributed to the scattering effects of ATAL aerosols, which mainly consist of sulfate and carbonaceous aerosols, as highlighted in their study. This clarification has been incorporated into the revised manuscript.*

L108 – Replace 'simulations with' to 'the'.

**Reply:** *We have revised the sentence accordingly in the revised manuscript.*

L111 – Why does the incoming solar radiation increase at the TOA?

**Reply:** *This statement refers to the positive radiative forcing of ATAL aerosols at the top of the atmosphere (TOA). Gao et al. (2023) demonstrated that ATAL aerosols increase the shortwave radiative flux at the TOA by approximately 0.15 W m⁻² over a diurnal cycle. This increase is primarily attributed to stronger aerosol absorption and re-emission, particularly during midday when solar insolation is highest due to the smaller solar zenith angle. In the revised manuscript, we have reworded this statement for improved clarity.*

L113 – Is there a citation for the changes in surface temperature?

**Reply:** *Yes. The statement regarding the potential reduction in surface temperature is based on Vernier et al. (2015). We have included this reference in the revised manuscript to support the discussion.*

L120 – research aspects -> research questions

**Reply:** *Removed in the revised manuscript while reframing the statement.*

L163 – What is the maximum altitude of the measurements?

**Reply:** *The average maximum altitudes reached at Gadanki, Hyderabad, and Varanasi are approximately 29 km, 27 km, and 30 km, respectively. To ensure the inclusion of UTLS aerosol signatures, we analyzed aerosol profiles up to a minimum altitude of 20 km. This clarification has been included in the revised manuscript.*

L175-191 – It is not easy to understand the differences in water vapor and ozone between different measurement locations. The exact values might not be needed here. I am also wondering why ozone and water vapor measurements are included in this work.

**Reply:** *We acknowledge that the original figure made it difficult to distinguish the water vapor and ozone measurements across different locations. To improve clarity, we have revised the figure by using a logarithmic scale for water vapor and focusing on the 12–20 km region.*

*Regarding their inclusion, atmospheric parameters such as pressure, temperature, water vapor density, and ozone density are essential inputs for radiative transfer calculations using the Santa Barbara DISORT Atmospheric Radiative Transfer (SBDART) model (e.g., Santhosh et al., 2024). While we do not explicitly analyze the radiative effects of water vapor and ozone on ATAL aerosols, these parameters are necessary for accurate radiative forcing estimations. To enhance readability, we have moved this discussion to the supplementary material.*

L192–199- This paragraph might not necessary here. Without including the details, I do not see much relevance to this work.

**Reply:** *We agree with the suggestion and have removed the paragraph in the revised manuscript.*

L231 – What is the cluster analysis? I think it is important to identify the types of aerosols over these locations and yet, this sentence is too brief to understand the processes.

**Reply:** *In the lower atmosphere (below 10 km), aerosol composition was characterized using seven-day air mass back trajectories at 500 m (boundary layer) and 4000 m (free troposphere)*

*above mean sea level, generated with the HYbrid Single Particle Lagrangian Integrated Trajectory (HYSPLIT) model (Stein et al., 2015). To classify distinct aerosol transport pathways, we applied hierarchical clustering using the Total Spatial Variance (TSV) method, optimizing the number of clusters based on minimal additional TSV reduction.*

*This analysis helps identify aerosol types by linking air mass origins, transport history, and residence time over source regions. A similar approach was used by Pawar et al. (2015) for aerosol classification through back trajectory analysis. After categorizing air-mass clusters based on potential aerosol types from the Optical Properties of Aerosols and Clouds (OPAC) database (Hess et al., 1998; Table S3), we obtained wavelength- and RH-dependent aerosol scattering properties (SSA and ASY) for radiative transfer calculations. A detailed description of this methodology is now included in Section S2 of the Supplementary Information.*

L232- What does a.m.s.l stand for?

**Reply:** *a.m.s.l refers to above mean sea level. This is expanded in the revised manuscript.*

L243 (methodology section, section 4.1) – Is the methodology used in this work is unique and has never been used before?

**Reply:** *Yes, the methodology used in this study for estimating the radiative forcing and heating rates of UTLS aerosols is unique and, to our knowledge, has not been applied in previous studies. While our recent work (Santhosh et al., 2025) employed an initial assumption of three dominant aerosol compositions—sulfate, nitrate, and anthropogenic aerosols—to define UTLS aerosol properties, the present study takes a more refined approach.*

*In the revised manuscript, we have replaced the original three-scenario framework with seven distinct aerosol composition scenarios (Table 3 in the revised manuscript) to better represent the complexity of UTLS aerosol mixtures. Additionally, unlike previous studies, our methodology integrates high-resolution in-situ balloon-borne observations with advanced radiative transfer calculations, incorporating a more comprehensive screening process for cloud contamination and aerosol type identification. These enhancements improve the accuracy of ATAL radiative forcing and heating rate estimates.*

L290-291- Please explain what ARF and HR are here.

**Reply:** *The terms ARF and HR refer to Aerosol Radiative Forcing and Heating Rates, respectively. We have clarified these abbreviations in the revised manuscript to ensure readability and avoid ambiguity.*

L306 –Does this assumption of three aerosol types (or scenarios) represent all the aerosol types?

**Reply:** *No. The original assumption of three aerosol scenarios does not fully represent all possible compositions of ATAL aerosols. While the anthropogenic aerosol scenario initially included a mixture of water-soluble (e.g., sulfate, nitrate, and organics) and water-insoluble*

*(e.g., mineral dust and hydrophilic organics) aerosols, it did not fully capture the complexity of ATAL composition. To address this limitation, we have replaced the anthropogenic scenario with external mixtures of sulfates, nitrates, organic carbon, and ammonium aerosols (Table 3 in the revised manuscript). These revised scenarios align more closely with reported UTLS aerosol compositions from Bossolasco et al. (2021) and Appel et al. (2022), ensuring a more comprehensive and representative assessment in the revised manuscript.*

L328- It is unclear what 'which is just beyond the sulfates' means.

**Reply:** *This statement was intended to clarify that the anthropogenic aerosol model used in the original manuscript (Continental Clean aerosol model from Hess et al., 1998) included sulfates, nitrates, organics, and other water-soluble substances in addition to the sulfate aerosols considered as the reference. To improve clarity and better represent realistic ATAL compositions, we have replaced this model with five distinct mixed aerosol compositions based on reported UTLS aerosol compositions. This ensures a more realistic depiction of ATAL aerosol characteristics in the revised manuscript.*

L368-What do the numbers in different colors in Fig. 5 mean?

**Reply:** *The numbers in different colors represent the percentage of air mass clusters reaching a given altitude from a specific direction. For example, a 50% cluster percentage indicates a 50% probability that the air mass arriving at the specified altitude over a given location originated from that particular direction. To enhance clarity, we have revised the figure caption and the corresponding text in the manuscript to provide an explicit explanation. Figure 5 in the original manuscript has been moved to the supplementary information as Figure S4.*

L431- 'from 13km to 19km'.

**Reply:** *Corrected in the revised manuscript.*

L434-It looks like the increase in aerosols is evident between 16-18 km.

**Reply:** *We agree with this observation and corrected it in the revised manuscript.*

L448-449-This suggests that the aerosol layer seems shallower than the 13-19 km rather than the complexity in my opinion.

**Reply:** *We acknowledge that the similar enhancement in aerosol backscatter ratios between 16 and 18 km across locations makes it challenging to establish a clear trend with latitude. Additionally, the vertical extent of these peaks varies across locations, contributing to the observed differences in elevation. To address this, we have revised the manuscript to avoid stating that the strength and complexity of ATAL aerosols increase from lower to higher latitudes. Since the peak backscatter ratio over Gadanki is slightly lower than at other locations, it suggests weaker aerosol enhancement at the edge of the ATAL region. However, additional observations, particularly from satellite measurements, are needed to confirm these patterns. This section has been revised accordingly in the manuscript.*

L450 (section 5.2)-This section suggests that the results in this study are consistent with previous studies. What is new and different in the results presented in this work that were not known previously?

**Reply:** *We appreciate the reviewer's comment. Section 5.2 of the original manuscript examines radiative forcing at the top of the atmosphere, at the surface, and within the atmospheric column under various UTLS aerosol scenarios while keeping aerosol properties unchanged at other altitudes. When we state that our results are "consistent" with previous studies, we mean that our radiative forcing estimates fall within the range of prior estimates, reinforcing their reliability. However, the cited studies differ in objectives and methodologies, and none specifically isolate the contribution of UTLS aerosols to columnar forcing.*

*To maintain the focus on UTLS forcing and avoid potential confusion, we have streamlined this section by removing comparisons that might divert attention from the study's core findings. The revised manuscript now emphasizes the novel aspects of our analysis, particularly the refined quantification of UTLS aerosol-induced radiative effects.*

L500-504 – What do the differences in the estimated radiative forcing mean? Are those significant differences?

**Reply:** *We appreciate the reviewer's comment. The differences in estimated radiative forcing (denoted as ΔARF in the original manuscript) refer to the change in aerosol radiative forcing relative to a reference composition, where sulfate aerosols dominate, representing the stratospheric background. In the revised manuscript, we have updated the notation from "ΔARF" to "δARF" to clarify that this represents the forcing difference due to aerosol composition rather than the forcing of a specific atmospheric layer. Mathematically,*

$$\delta ARFx = ARFx - ARF_{SUL}$$

*where x represents any of the alternative aerosol compositions considered in this study (nitrate-dominant (NIT), C1, C2, C3, C4, and C5). The δARF values indicate whether a given aerosol composition leads to net warming or cooling (depending on the sign) in the UTLS and the atmospheric column when deviating from the sulfate-dominant reference (SUL).*

*These differences are significant within the UTLS. After revising the results, we found that δARF ranges from 0.06 to 0.28 W m⁻², which corresponds to a 3- to 14-fold increase in radiative forcing compared to the reference conditions. This substantial variation underscores the strong sensitivity of radiative forcing to aerosol composition in the UTLS, indicating that even minor changes in aerosol properties or concentrations can have disproportionately large impacts on the radiative balance of this region.*

L526 – Need a reference for the radiative forcing from increase $CO_2$.

**Reply:** *This refers to the result from Vernier et al. (2015), who reported a clear-sky radiative*

*forcing of -0.12 W m$^{-2}$ at the top of the atmosphere due to ATAL, which partially offset the global radiative forcing from increased $CO_2$ (0.3 W m$^{-2}$). The reference for the forcing due to increased $CO_2$ is from Solomon et al. (2011). As this comparison may divert readers' attention to global scenarios and the greenhouse gas forcing, which are beyond the scope of the present study, we have removed this statement from the revised manuscript. However, the study by Vernier et al. (2015) is one of the earliest analyses of the ATAL radiative forcing. Hence, we have provided this information in the introduction section of the revised manuscript.*

L551-552 – A reference is need for the -2-32 Wm$^{-2}$ surface forcing per unit AOD here.

**Reply:** *This inference is based on our estimates. We calculated the surface aerosol radiative forcing per unit AOD by dividing $\Delta ARF_{SUR}$ values (-0.01 W m$^{-2}$ to -0.16 W m$^{-2}$) by the UTLS aerosol optical depth ($AOD_{UTLS}$ at 500 nm = 0.005), yielding a range of -2 to -32 W m$^{-2}$ per unit AOD at 500 nm. However, as these results shift the focus from the UTLS region to surface forcing, we have removed them from the revised manuscript to maintain alignment with the study's core objectives.*

L569 – Citation is needed after "various researchers".

**Reply:** *We acknowledge the reviewer's suggestion for proper citation. Several studies have used sulfate aerosol properties as a proxy to investigate the radiative impacts of nitrate aerosols, including Wang et al. (2010), Liao et al. (2004), and Van Dorland et al. (1997). As discussed in our recent study (Santhosh et al., 2025), which highlights the importance of considering nitrate optical properties instead of sulfate optical properties, we have provided a citation to this work, and the original statement has been removed from our revised manuscript.*

L574-577- It is not clear what the message is in these sentences.

**Reply:** *The original sentences aim to highlight that nitrate aerosol optical properties should not be substituted with sulfate properties, as our findings indicate negligible differences in shortwave radiative forcing and heating rates under completely nitrate- and sulfate-dominant UTLS compositions. While SSA and ASY values for these aerosols are nearly identical at shorter wavelengths, differences become noticeable at longer wavelengths. For example, at approximately 2.8 μm and relative humidity (RH) below 40%, the single-scattering albedo (SSA) for nitrates is approximately 40% higher than that for sulfates (Zhang et al., 2012), indicating stronger absorption by nitrate aerosols at these wavelengths.*

*This highlights the importance of higher-wavelength measurements in accurately capturing the radiative impacts of nitrate aerosols. Since this aspect is already discussed in Santhosh et al. (2025), we have cited that study in the revised manuscript to avoid redundancy while maintaining clarity and conciseness.*

L624-631-Are ATAL and EALs separate phenomena? I am not sure what the purpose of introducing EALs here is.

**Reply:** *No. The Asian Tropopause Aerosol Layer (ATAL) is also considered an Elevated Aerosol Layer (EAL) observed in the upper troposphere and lower stratosphere (UTLS) region, unlike those in the free troposphere, which is primarily linked to the Asian summer monsoon. In the original manuscript, the discussion in lines 624–631 referred to EALs in the free troposphere during the pre-monsoon and monsoon periods. However, since this study does not explicitly analyze free-tropospheric EALs, we have removed this section from the revised manuscript to maintain focus and clarity, as suggested by the reviewer.*

*We sincerely thank the reviewer once again for suggesting valuable solutions that significantly enhanced the content of the manuscript.*

**References**

Appel, O., Köllner, F., Dragoneas, A., Hünig, A., Molleker, S., Schlager, H., Mahnke, C., Weigel, R., Port, M., Schulz, C., Drewnick, F., Vogel, B., Stroh, F., and Borrmann, S.: Chemical analysis of the Asian tropopause aerosol layer (ATAL) with emphasis on secondary aerosol particles using aircraft-based in situ aerosol mass spectrometry, Atmos. Chem. Phys., 22(20), 13607–13630, https://doi.org/10.5194/acp-22-13607-2022, 2022.

Bossolasco, A., Jegou, F., Sellitto, P., Berthet, G., Kloss, C., and Legras, B.: Global modeling studies of composition and decadal trends of the Asian Tropopause Aerosol Layer, Atmos. Chem. Phys., 21(4), 2745–2764, https://doi.org/10.5194/acp-21-2745-2021, 2021.

Draxler, R. R., and Hess, G. D.: An overview of the HYSPLIT_4 modelling system for trajectories, dispersion and deposition, Aus. Meteorol. Mag., 47(4), 295–308, 1998.

Fadnavis, S., Semeniuk, K., Pozzoli, L., Schultz, M. G., Ghude, S. D., Das, S., and Kakatkar, R.: Transport of aerosols into the UTLS and their impact on the asian monsoon region as seen in a global model simulation, Atmos. Chem. Phys., 13(17), 8771–8786, https://doi.org/10.5194/acp-13-8771-2013, 2013.

Gao, J., Huang, Y., Peng, Y., and Wright, J. S.: Aerosol Effects on Clear-Sky Shortwave Heating in the Asian Monsoon Tropopause Layer, J. Geophys. Res.-Atmos., 128(4), 1–23, https://doi.org/10.1029/2022JD036956, 2023.

Hess, M., Koepke, P., and Schult, I.: Optical Properties of Aerosols and Clouds: The Software Package OPAC, Bull. Am. Meteorol. Soc., 79(5), 831–844, https://doi.org/10.1175/1520-0477(1998)079<0831:OPOAAC>2.0.CO;2, 1998.

Liao, H., Seinfeld, J. H., Adams, P. J., and Mickley, L. J.: Global radiative forcing of coupled tropospheric ozone and aerosols in a unified general circulation model, J. Geophys. Res., **109**, D16207, https://doi.org/10.1029/2003JD004456, 2004.

Ma, J., Brühl, C., He, Q., Steil, B., Karydis, V. A., Klingmüller, K., Tost, H., Chen, B., Jin, Y., Liu, N., Xu, X., Yan, P., Zhou, X., Abdelrahman, K., Pozzer, A., and Lelieveld, J.: Modeling the aerosol chemical composition of the tropopause over the Tibetan Plateau during the Asian summer monsoon, Atmos. Chem. Phys., 19(17), 11587–11612, https://doi.org/10.5194/acp-19-11587-2019, 2019.

Santhosh, V. N., Madhavan, B. L., and Venkat Ratnam, M.: Quantifying shortwave radiative forcing and heating rates of UTLS aerosols in the Asian summer monsoon anticyclone region, J. Quant. Spectrosc. Ra., 339, 109430,

*https://doi.org/10.1016/j.jqsrt.2025.109430*, 2025.

Santhosh, V. N., Madhavan, B. L., Ratnam, M. V., Naik, D. N., and Sellitto, P.: *Assessing biases in atmospheric parameters for radiative effects estimation in tropical regions, J. Quant. Spectrosc. Radiat. Transf., 314, 108858, https://doi.org/10.1016/j.jqsrt.2023.108858, 2024a.*

Stein, A. F., Draxler, R. R., Rolph, G. D., Stunder, B. J. B., Cohen, M. D., and Ngan, F.: *NOAA's hysplit atmospheric transport and dispersion modeling system, Bull. Amer. Meteor. Soc., 96(12), 2059–2077, https://doi.org/10.1175/BAMS-D-14-00110.1, 2015.*

Van Dorland, R., Dentener, F. J., and Lelieveld, J.: *Radiative forcing due to tropospheric ozone and sulfate aerosols, J. Geophys. Res., **102**, D23, 28079–28100, 1997.*

Vernier, J.-P., T. D. Fairlie, M. Natarajan, F. G. Wienhold, J. Bian, B. G. Martinsson, S. Crumeyrolle, L. W. Thomason, and K. M. Bedka: *Increase in upper tropospheric and lower stratospheric aerosol levels and its potential connection with Asian pollution, J. Geophys. Res.-Atmos., 120, 1608–1619, https://doi.org/10.1002/2014JD022372, 2015.*

Wang, T. J., Li, S., Shen, Y., Deng, J. J., and Xie, M.: *Investigations on direct and indirect effect of nitrate on temperature and precipitation in China using a regional climate chemistry model system, J. Geophys. Res., 115, D00K26, https://doi.org/10.1029/2009JD013264, 2010.*

Zhang, H., Shen, Z., Wei, X., Zhang, M., and Li, Z.: *Comparison of optical properties of nitrate and sulfate aerosol and the direct radiative forcing due to nitrate in China, Atmos. Res., 113, 113–125, https://doi.org/10.1016/j.atmosres.2012.04.020, 2012.*